# Surface and Interface Designs in Copper-Based Conductive Inks for Printed/Flexible Electronics

**DOI:** 10.3390/nano10091689

**Published:** 2020-08-27

**Authors:** Daisuke Tomotoshi, Hideya Kawasaki

**Affiliations:** Department of Chemistry and Materials Engineering, Faculty of Chemistry, Materials and Bioengineering, Kansai University, Suita-shi, Osaka 564-8680, Japan; k432580@kansai-u.ac.jp

**Keywords:** copper, nanoparticles, complexes, inks, pastes, printed electronics, flexible devices

## Abstract

Silver (Ag), gold (Au), and copper (Cu) have been utilized as metals for fabricating metal-based inks/pastes for printed/flexible electronics. Among them, Cu is the most promising candidate for metal-based inks/pastes. Cu has high intrinsic electrical/thermal conductivity, which is more cost-effective and abundant, as compared to Ag. Moreover, the migration tendency of Cu is less than that of Ag. Thus, recently, Cu-based inks/pastes have gained increasing attention as conductive inks/pastes for printed/flexible electronics. However, the disadvantages of Cu-based inks/pastes are their instability against oxidation under an ambient condition and tendency to form insulating layers of Cu oxide, such as cuprous oxide (Cu_2_O) and cupric oxide (CuO). The formation of the Cu oxidation causes a low conductivity in sintered Cu films and interferes with the sintering of Cu particles. In this review, we summarize the surface and interface designs for Cu-based conductive inks/pastes, in which the strategies for the oxidation resistance of Cu and low-temperature sintering are applied to produce highly conductive Cu patterns/electrodes on flexible substrates. First, we classify the Cu-based inks/pastes and briefly describe the surface oxidation behaviors of Cu. Next, we describe various surface control approaches for Cu-based inks/pastes to achieve both the oxidation resistance and low-temperature sintering to produce highly conductive Cu patterns/electrodes on flexible substrates. These surface control approaches include surface designs by polymers, small ligands, core-shell structures, and surface activation. Recently developed Cu-based mixed inks/pastes are also described, and the synergy effect in the mixed inks/pastes offers improved performances compared with the single use of each component. Finally, we offer our perspectives on Cu-based inks/pastes for future efforts.

## 1. Introduction

Copper (Cu) is one of the significant metals in the industry that can be used in various applications. It has several excellent properties, including electrical conductivity, thermal conductivity, and antimicrobial activity [1]. An alloy can be fabricated using Cu in combination with other elements, such as gold (Au), silver (Ag), aluminum (Al), nickel (Ni), tin (Sn), and zinc (Zn), to improve its properties, such as corrosion resistance or machinability. Besides, the ease of fabrication and the cost-effectivity of Cu are also important in industrial applications. The high electrical conductivity (i.e., low resistivity, ~1.68 μΩ·cm) is the most attractive characteristic of Cu as an electronic material [1]. For the last 10 years, novel electronic forms have been developed from their traditional rigid/bulky forms to soft/thin forms which are flexible, bendable, and stretchable. The softness is attributed to the use of flexible substrates, such as polymers (polyethylene terephthalate (PET), polyethylene naphthalate (PEN), polyimide (PI), and polycarbonate (PC)), paper, textiles, and rubber. Recent printing technologies, including flexography, offset, gravure, screen printing, and inkjet printing, offer the direct deposition of conductive materials on flexible substrates for the cost-effective/large-scale fabrication of flexible electronics [2]. Printed electronics are essential to facilitate the widespread use of flexible electronics and, more recently, stretchable electronics, such as sensors, electronic displays, solar cells, thin-film transistors, and supercapacitors [3,4,5,6,7,8,9,10,11,12,13,14,15,16,17,18,19,20,21]. Printable conductive electrodes on flexible substrates require the following characteristics for the application of printed electronics: (i) a high conductivity, (ii) cost-effectiveness, (iii) durability under extreme environmental conditions (heat and moisture), and (iv) printing processability on low-cost plastic substrates (indispensably, PET for productive roll to roll (R2R) processes).

In the printing process, conductive materials dispersed in solvents (i.e., ink/paste) are deposited on flexible substrates in predesigned patterns. Numerous types of conductive inks/pastes have been developed by using various conductive sources, such as graphene, carbon (C) nanotubes, metal (Ag, Au, and Cu) nanoparticles/nanowires, metal oxide (Zn, Al, and Sn) nanoparticles, conducting polymers, and organic-metal complexes, for different applications [22,23,24]. These conductive materials have advantages and disadvantages for conductive inks/pastes. Conductive organic inks/pastes (i.e., polymers and carbon) are suitable for flexible electronics owing to their natural softness. Moreover, they are well compatible with polymer substrates. However, their electrical conductivities are not ideal for electronic applications. Metal-based inks/pastes have advantages in terms of their high electrical/thermal conductivities. However, their flexibility and extensibility are not sufficient to fabricate flexible and stretchable devices. Besides, the binding of metals to polymer substrates is weak without the use of an adhesive. Solving such problems remains challenging for conductive inks/pastes in printed/flexible electronics.

Ag has been utilized as a metal for fabricating metal-based inks/pastes for printed/flexible electronics. Ag has a high electrical conductivity among various metals, and it is relatively cost-effective compared with Au. The oxidation of Ag occurs slowly, slightly affecting the conductivity of Ag-based inks/pastes over time. Besides, Ag nanoparticle-based inks can produce conductive films at low-temperature sintering (less than 150 °C), and this is the advantage of the formation of conductive patterns on flexible substrates with a low heat resistance. For these reasons, Ag-based inks/pastes have been frequently used as metal-based inks/pastes and commercialized by numerous companies. The current challenge is the replacement of Ag by other cheaper conductive metals, such as Cu, Al, or Ni. Among these metals, Cu is the most promising candidate for metal-based inks/pastes [25]. Cu has a high intrinsic electrical/thermal conductivity (similar to Ag and higher than Au), and it is more cost-effective and abundant (about 1% of the price of Ag and 1000 times abundant than Ag). More importantly, the migration tendency of Cu is less than that of Ag. Recently, therefore, the Cu-based inks/pastes for printed/flexible electronics have gained increasing attention. However, the disadvantages of Cu-based inks/pastes include their instability against oxidation under an ambient condition and their tendency to form insulating layers of Cu oxide, such as cuprous oxide (Cu_2_O) and cupric oxide (CuO). The formation of the oxidation layer causes a low conductivity of sintered films and interference in the sintering process of Cu particles.

In the past decade, numerous researchers have developed stable Cu-based inks/pastes. Many excellent review articles on the widespread use of flexible/stretchable electronics have been reported [3,4,5,6,7,8,9,10,11,12,13,14,15,16,17,18,19,20,21], as well as on Cu-based conductive inks/pastes [26,27,28]. Specifically, we adopt the viewpoints of the surface and interface designs for Cu-based inks/pastes in this review, as summarized in Figure 1. The ability to integrate the oxidation resistance and low-temperature sintering into Cu-based inks/pastes is determined by the surface and interface designs. We mainly describe the recent progress of Cu-based inks/pastes in the past 5 years (i.e., since 2015) from the perspective of surface and interface designs. In Section 2, we categorize various types of Cu-based inks. In Section 3, we briefly describe the surface oxidation behaviors of Cu. In Section 4 and Section 5, we discuss various approaches on the surface designs for Cu-based inks/pastes to achieve both oxidation resistance and low-temperature sintering to produce highly conductive Cu patterns/electrodes on flexible substrates. These surface control approaches include surface designs by polymers, small ligands, core-shell structures, and surface activation. In Section 6, Cu-based mixed inks/pastes are described, including Cu nanoparticles/microparticles, metal-organic decomposition (MOD)/Cu particles, Cu nanowires/Cu nanoparticles, multiwalled carbon nanotube (MWCNT)/Cu nanoparticles, and Cu nanoparticles/other metal nanoparticles. The synergistic effect of the mixed inks/pastes is described to improve the performance compared with the single use of each component. In Section 7, we describe the adhesion enhancement of sintered Cu films on flexible substrates using adhesion promoters and the surface modification of flexible substrates. Finally, we offer the remaining challenges and our perspectives for future studies on Cu-based inks/pastes for printed/flexible electronics.

## 2. Categories of Cu-Based Inks/Pastes

The main components of Cu-based inks/pastes are as follows:Cu particles or their precursors (e.g., Cu salts or Cu formate-amine complex) as conductive fillers.Solvents for the stable dispersion of Cu particles or dissolution of their precursors, facilitation of transport/drying of the ink/pastes, and control of viscosity/surface tension for printing. Besides, reductive solvents, such as ethylene glycol (EG), can be used for Cu-based inks/pastes to suppress Cu oxidation at high temperatures.Dispersion and protective agents for the dispersion stability of Cu particles and the Cu oxidation resistance.Other additives to enhance or modify the properties of the ink/paste (e.g., antioxidants, binders, and adhesion promoters).

We can divide Cu-based inks/pastes into four categories from the viewpoints of Cu sources (Figure 2): (i) traditional micron-sized flake/powder type, (ii) nanoparticle type, (iii) nanowire type, and (iv) precursor ion type [28].

The sintering temperature ranges from 50 to 80% of the melting temperature of the material, but for nanosized powders, the sintering temperature can be much lower, i.e., 20 to 30% of the melting temperature [29]. Thus, the application of micron-sized powder-based pastes is limited to low-temperature sintering. Compared with micron-sized Cu powders, nanoparticle-based inks have a much lower sintering temperature (<250 °C). When the particle size of a metal reaches the nanoscale, its melting point decreases [30], thus resulting in the reduction in the sintering temperature of the Cu nanoparticles. The reduction in the total interfacial energy causes this sintering process of Cu nanoparticles via the densification–reduction in the solid/liquid interfacial area. Using Cu nanoparticle-based inks, we can obtain sintered Cu films at low-temperature (<150 °C) sintering on the thermal-sensitive flexible substrates (e.g., polymer, paper, textile, and rubber). The Cu precursor ink (category iv) is also useful for reducing the process temperature. Cu precursors are metals in oxidized forms, which can be organic Cu complexes, such as Cu formate complexes with various alkylamines. During the thermal process, the Cu precursor is decomposed and causes the in-site formation of Cu nanoparticles at temperatures less than <150 °C, resulting in the conductive Cu film on flexible substrates.

## 3. Surface Oxidation Behavior of Cu Nanoparticles

The surface oxide layer on Cu nanoparticles exerts two adverse effects: (i) it increases the sintering temperature and (ii) reduces electrical conductivity. Therefore, controlling the surface oxidation on Cu nanoparticles is important to minimize surface oxidation [31,32,33,34,35,36,37,38,39]. Yabuki et al. investigated the oxidation behavior of Cu nanoparticles (~20 nm in size) under a 20% oxygen-nitrogen (N_2_) condition by the mass gain of Cu nanoparticles at 150–300 °C via thermogravimetry (TG) [31]. They found that a threshold temperature for the mass increase in the oxidation was recognized between 190 and 200 °C. Below the threshold temperature, Cu nanoparticles could only be changed to Cu_2_O at the first stage. Conversely, above the threshold temperature, an initial and drastic Cu_2_O formation was observed, which then changed to CuO on the surface of the particles. The mass fractions of Cu, Cu_2_O, and CuO in oxidized nanoparticles obtained at 170 °C after 200 min annealing were 80% Cu_2_O and 20% CuO, while they were 10% Cu_2_O and 90% CuO at 240 °C (Figure 3a,b). This indicates that the mass fraction of Cu_2_O is more dominant in lower temperatures.

Jeong et al. reported the effect of the molecular weight of polyvinyl pyrrolidone (PVP) protective layers on the surface oxide layer of Cu nanoparticles [32]. The thickness of the surface oxide layer decreases with an increasing PVP molecular weight, from an average of 3.1 to 1.6 nm as the PVP molecular weight is increased from 10,000 to 40,000 g mol^−1^. PVP with a higher molecular weight is more densely packed on the Cu surface, thus inhibiting the access of the oxygen atoms to the Cu surface (i.e., reduced surface oxidation). Jeong et al. also demonstrated that the electrical resistivities of Cu films decrease with the decreasing thickness of the surface oxide layer. This indicates that the thin surface oxide layer is important for producing low-resistivity Cu films.

Glaria et al. reported the effect of hexadecylamine (HDA) or tetradecyl-phosphonic (TDPA) ligands on the oxidation rate of colloidal Cu nanoparticles [33]. Metallic Cu nanoparticles have a localized surface plasmon resonance (LSPR) peak at around 550 nm, and the surface oxidation makes the peak obscure. The LSPR peak of Cu nanoparticles in real-time UV–Vis spectra was employed to determine the extent of conversion from metallic Cu to Cu oxides which do not exhibit the LSPR band. HDA ligands were more strongly coordinated on surface Cu sites owing to the high density of the hydrophobic alkylamine shell and the chemical nature of the amine ligand (neutral two-electron donor), thus allowing an extremely efficient oxidation protection of the Cu core. However, the TDPA ligands caused a rapid increase in the oxidation rate of Cu nanoparticles and, finally, a re-dissolution of Cu(II) species as a molecular complex in the solution. This indicates the efficient protection of the amine ligands of Cu nanoparticles compared with phosphonic ligands.

Moreover, Cure et al. found that the oxidation rate of HDA-protected Cu nanoparticles has three distinct regimes [34]: (i) peroxide island formation in a fast regime via the adsorption/desorption of oxygen on the nanoparticle; (ii) coalescence of the oxide islands in a slower regime to form a complete Cu_2_O shell; (iii) extremely slow oxidation of the residual Cu core and the formation of hollow Cu_2_O nanoparticles (the so-called nanoscale Kirkendall effect). It is suggested that the outward and unbalanced diffusion of Cu atoms is attributed to the Kirkendall effect, resulting in Cu vacancies in the nanoparticle core. Further coalescence of the vacancies eventually creates a void at the center of the particle [35,36]. The oxidation rate of Cu nanoparticles can be reduced by adding HDA ligands due to the competitive adsorption between O_2_ and HDA on the Cu nanoparticle [34]. The period when the Cu_2_O shell is completely formed was shifted from 1 h to 20 days by increasing the HDA content from 0.1 to 2 eq [34].

Pacioni et al. found another insight into the oxidation behavior of unprotected Cu nanoparticles in water, which were photochemically synthesized from aqueous Cu sulfate and I-2959 [37]. The unprotected Cu nanoparticles in water were easily oxidized in the air to yield aqueous Cu^2+^ rather than Cu oxides. They proposed that residual Cu^+^ reacts with O_2_ to give the unstable adduct CuO_2_^+^ that decomposes rapidly to yield Cu^2+^ and O_2_^−^ (or HO_2_^·^). The oxidation could be inhibited by quenching O_2_^−·^ (or HO_2_^·^) with adding L-ascorbic acid or keeping it at a low temperature of 5 °C.

Aside from alkylamines, Tang et al. found useful polyamines of polyethylenimine (PEI) for the oxidation resistance of Cu nanoparticles in aqueous solution [38]. The three typical moieties of PEI, i.e., primary amine, secondary amine, and tertiary amine, form strong chemical bonds with the Cu nanoparticle surface through the coordination between Cu and the amines. The adsorption energy between the amines and Cu(111) surface was calculated using the quantum mechanical modeling method (density-functional theory, DFT). The estimated adsorption energies of primary amine, secondary amine, and tertiary amine groups are −0.561, −0.549, and −0.294 eV, respectively, indicating the increased strength of chemisorption in the order of primary amine > secondary amine > tertiary amine groups (Figure 3c–e).

Dabera et al. reported the unexpected findings of the correlation between the air stability of Cu nanoparticles and the structure of different thiolate-protected ligands [39]. They prepared eight thiolate-protected Cu nanoparticles on a substrate by the ligand exchange from oleylamine (OAM)-protected Cu nanoparticles without etching the Cu core (Figure 3f). It is not always true that a longer ligand length prevents Cu oxidation. Shorter alkyl thiol ligands, such as 3-mercaptopropionic acid, offered the highest stability against the oxidation, compared with the cases of longer ones. An alkyl thiol ligand terminated with hydrophilic end groups (–COOH) is proposed to determine the effectiveness of the ligand in blocking the oxidation of Cu nanoparticles (Figure 3g).

## 4. Surface Designs by Surface Protective Layers Against the Cu Oxidation in Cu-Based Inks/Pastes

The challenging aspect of Cu-based inks/pastes is the oxidation of Cu in the air to form Cu oxide, especially at high temperatures. The Cu oxide is an electrically insulating material which reduces Cu film conductivity. Moreover, it suppresses the sintering of Cu nanoparticles. Thus, the thermal sintering process of Cu-based inks/pastes has been performed under oxygen-free or reductive environments to prevent significant oxidation. The oxidation tendency of Cu is more enhanced as the size of Cu nanoparticles becomes smaller. Therefore, smaller-sized Cu nanoparticles exert a positive effect on low sintering temperatures but a negative effect on easy oxidation. The surface of Cu nanoparticles is generally covered with protective agents of organic ligands or polymers to prevent Cu nanoparticle oxidation. These protective agents are mainly divided into two categories: organic protective agents (e.g., polymers and small organic ligands) and inorganic layers (e.g., metal shell and C shell). The protective agents can reduce agglomeration in the Cu nanoparticles and prevent oxidation of the Cu nanoparticles. However, the protective agents need to be thermally removed via the sintering process for a better sintering of the Cu nanoparticles in the inks/pastes. Thus, protective agents for Cu-based inks/pastes need capabilities of preventing the aggregation/oxidation of Cu nanoparticles and thermal removal during the sintering process. 

Solution-based synthetic methods of Cu nanoparticles can be mainly divided into the following categories: chemical reduction, microemulsion, thermal decomposition, biological synthesis, and electrochemical synthesis [25,40,41,42]. Among them, chemical reduction methods are mostly utilized to synthesize Cu nanoparticles for Cu-based inks/pastes owing to their adjustable size and easy synthesis of small Cu nanoparticles. In the chemical reduction synthesis, Cu nanoparticles are obtained by reducing Cu salts (e.g., CuSO_4_, Cu(II) acetylacetonate, CuCl_2_, or Cu(NO_3_)_2_) with reducing agents (e.g., sodium borohydride, hydrazine, ascorbic acid (AA), glucose, or NaH_2_PO_2_) in the presence of protective agents [25,40,41,42]. It should be noted that the process involving product purification to obtain a good quality of the final Cu nanoparticles takes time and effort. For example, multiple washing operations require as-prepared colloidal Cu nanoparticles to remove the excess protective agents, the unreacted reagents, and the adsorbed inorganic salts generated in the reaction, as they would often increase the electrical resistivity of the final conductive inks. In this section, we will mainly describe the surface designs with the protective agents for Cu-based inks/pastes. 

### 4.1. Surface Designs by Polymers

The surface designs by polymers for Cu-based inks/pastes are summarized in Table 1.

#### 4.1.1. Polyvinyl Pyrrolidone (PVP)

Various protective agents have been utilized to control the particle’s growth and stabilize Cu nanoparticles against aggregation/oxidation. Among these agents, PVP is widely used for the chemical reduction synthesis of Cu nanoparticles and other metal nanoparticles [43]. PVP has the ability to coordinate and interact with both cupric ion and Cu nanoparticles by lone pairs of electrons on the oxygen atoms of the PVP unit. This prevents aggregation/oxidation during the synthesis. Hirai et al. reported polymer-stabilized Cu nanoparticles using various polymer protective agents, such as PVP, poly(vinyl alcohol), poly(methyl vinyl ether), (2-hydroxyethyl) cellulose, poly(acrylic acid), poly(2-acrylamide-2-methyl-1-propane sulfonic acid), and the copolymer of methyl vinyl ether and maleic acid [44]. The use of PVP was most effective for the synthesis of plasmonic Cu nanoparticles with a high oxidation resistance.

Granata et al. synthesized PVP-protected Cu nanoparticles via the aqueous reduction in CuSO_4_ with D-glucose for the Cu nanoparticle-based inks. The increase in the cupric concentration in the range of 0.1–1.0 M resulted in a progressive increase in the size of Cu nanoparticles from 30 to 200 nm. The electrical resistivity of the Cu-based conductive ink of 92-nm Cu nanoparticles was 36 μΩ cm at a temperature of 300 °C under an air–formic acid (95–5%) environment [45]. Cheng et al. synthesized PVP-protected Cu nanoparticles with a size of 140 nm in diethylene glycol at 90 °C by reducing Cu hydroxide with L-ascorbic acid [46]. The Cu nanoparticles were stable against oxidation and could be stored for 3 months. The protective layer consisting of PVP and L-ascorbic acid on the particles contributed to the antioxidant performance. The sintered Cu film exhibited low resistivities of 17 and 5.7 μΩ·cm at the sintering temperature of 250 and 400 °C, respectively, under an Ar (96%)–H_2_ (4%) gas mixture.

Liu et al. proposed a modified polyol synthesis of PVP-protected Cu nanoparticles with a small size of 30 nm in diethylene glycol at 90 °C by reducing CuSO_4_ with sodium phosphinate [47]. The small Cu nanoparticles have the native amorphous oxide layer on the surface of the as-prepared Cu nanoparticles. Liu et al. removed the native amorphous layer through formic acid treatment, leaving the formate attached to the Cu nanoparticle surface, which could decompose at low temperatures. The formic acid-treated Cu nanoparticle (f-Cu NPs) films sintered at 260 °C under an N_2_(95%)–H_2_(5%) gas mixture exhibited a low electrical resistivity of 6.1 μΩ·cm, which was much lower than that (18.6 μΩ·cm) of the sintered Cu film without formic acid treatment (u-CuNPs) due to the enhanced sintering neck on the f-Cu NPs. 

Lee et al. found that the effects of sodium dodecylbenzene sulfonate (SDBS) prevented the oxidation of sintered Cu circuits under high-temperature storage (HTS) and a steady-state temperature humidity bias life test (HTHH) under 85 °C/85% relative humidity environmental conditions [48]. The Cu-based ink from PVP-protected Cu nanoparticles (about 40 nm in size) was formulated for screen printing and improving the adhesion with the substrate, including ethylcellulose and various contents of SDBS (0, 1, 2 and 3 wt% per paste), and α-terpineol as the paste solvent. The microstructures of sintered Cu circuits (first step, 250 °C for 30 min in the air; second step, 250 °C for 30 min in H_2_) exhibited interconnected structures through necking, and the printed Cu circuits with SDBS exhibited denser microstructures. After 24, 48, and 72 h of HTS and HTHH tests, the sintered Cu circuits with SDBS suppressed the increases in electrical resistivity. The X-ray diffraction (XRD) patterns showed that Cu oxidation decreased in the presence of SDBS.

The main bottleneck of PVP-protected Cu nanoparticle ink is the typical sintering temperature higher than 250 °C, since the thermal decomposition of PVP can occur above 250 °C and the hydrogen-driven reduction temperature should be higher than 230 °C to eliminate the surface oxide layers [45,46,47,48]. The high sintering temperature limits the use of flexible substrates, such as PET or PC, because of the low softening point (Tg) below 150 °C. In addition, the inert/reduced gas environment would cancel the low-cost advantage of using Cu nanomaterials and high-temperature thermal sintering processes. To solve this problem, Woo et al. utilized various carboxylic acids (formic, acetic, propionic, and butyric acids) as reducing agents during the thermal sintering of PVP-protected Cu nanoparticles in inks [49]. The carboxylic acids were exposed to the Cu film by N_2_ gas that was bubbled through the liquid acids during the annealing process. The sintered Cu films exposed to formic acid vapor exhibited the lowest resistivity of 3.1 μΩ cm (comparable to bulk Cu resistivity) when annealed at 200 °C for 60 min. In addition, the oxalic acid reduced Cu oxide compared with formic acid with a shortening of the heat time to produce a highly conductive Cu film. The role of carboxylic acid was proposed as follows: the carboxylic acids with shorter hydrocarbon chains are stronger acids due to the lower pKa, allowing for more dissociated carboxylate ions. Thus, the short-carboxylic acids readily react with the surface CuO to form Cu carboxylates that are reduced to metallic Cu in the annealing process [49].

Intense pulsed light (IPL) sintering, such as the use of a white flashlight from a xenon lamp, has been employed for PVP-protected Cu-based inks. In IPL sintering, the Cu nanoparticles can absorb the flashlight ranging in the UV–Vis wavelength, and the photon energy is instantly (a few milliseconds) irradiated from a flash lamp as a thermal heat source, resulting in short-time sintering. Another advantage of IPL is the conversion of the PVP chains to intermediate alcohols and acids by flash white light irradiation, followed by the reduction in the Cu oxide shell through the photo-induced intermediates. As a result, a highly conductive Cu film with a few μΩ cm can be obtained under ambient conditions without causing damage to the flexible substrates employing optimized IPL sintering conditions (e.g., pulse duration, irradiation energy, and number of pulses). Such reactive sintering of PVP-protected Cu nanoparticles using the IPL methods has been widely adopted in the sintering of Cu-based inks/pastes for printed/flexible electronics [50]. Apart from IPL sintering, laser sintering and plasma sintering are also developed for PVP-protected Cu nanoparticle inks. Herein, we discuss the results on recent IPL, laser, and plasma sintering methods using PVP-protected Cu nanoparticle inks.

Kim’s group has developed deep UV/near-infrared (NIR)-assisted IPL sintering (irradiation energy: 12.5 J/cm^2^, pulse duration: 10 ms, pulse number) [51]. The combination of deep UV irradiation and flash white light irradiation on PVP-protected Cu nanoparticle inks reduced the resistivity of the sintered Cu films, which have the lowest resistivity (7.62 μΩ·cm) (Figure 4a,b). PVP is decomposed very effectively by deep UV (198 nm) irradiation owing to its UV absorption characteristics. They further investigated the effect of the wavelength used in the flash white light sintering on the resistivity and microstructure of the Cu films [52]. Using a band pass-filtered flashlight from 500 to 600 nm, which included the plasmonic absorption wavelength of the Cu nanoparticles (590 nm), the sintered Cu films exhibited denser necking connections of Cu nanoparticles and a lower porosity, resulting in the low resistivity (6.97 μΩ·cm).

To avoid serious crack formation on the sintered Cu films in IPL process, Chan et al. employed a progressive three-step sintering for PVP-protected Cu nanoparticle inks, involving the sequential processes of low-pressure drying, NIR sintering, and IPL reduction [53]. The three-step method improved the compactness of nanoparticles, the completeness of nanoparticle sintering, and the effectiveness of CuO reduction, respectively. As a result, the highly conductive Cu film (7.9 μΩ·cm) was obtained as a result of a combination of 65.7% surface roughness improvement, 37.3% porosity reduction, and 91.7% oxygen elimination. Additionally, this method also reduced the problematic delamination of the Cu film from a PI substrate in IPL-only sintering via the enhanced adhesion between the Cu film and the substrate in the three-step method.

The advantage of the laser sintering method is the selective annealing feature for in situ direct writing/patterning and fast working speed under an ambient condition without oxidation. Kwon et al. reported an oxidation-free selective laser (532-nm continuous-wave Nd/YAG visible laser) sintering of PVP-protected Cu nanoparticle inks in the air environment [54]. Various laser powers and laser scanning speeds were optimized for the Cu inks. The laser-sintered Cu film exhibited excellent electrical properties (<1 Ω/sq.) compared with those (2.01 × 10^4^ Ω/sq.) obtained through the conventional thermal heating process at 400 °C under an Ar environment. It was suggested that the degraded PVP during laser irradiation provides hydrogen species and acts as a reducing agent. The Cu inks deposited on PEN films are patterned by the selective laser sintering process to produce a flexible and transparent conductor grid pattern on a PEN film with a 20 μm line width. Moreover, the cyclic bending test with a radius of 2.25 mm up to 1000 cycles revealed the good mechanical stability of sintered Cu film on PEN with ±10% electrical resistance changing rate.

Jung et al. fabricated flexible micro/nano metal grid transparent conductors with extremely high figures of merit, FOM = 2000, and very narrow linewidth as low as 2.4 μm from PVP-protected Cu nanoparticle inks by thermal conduction layer (TCL)-assisted laser sintering [55]. The TCL worked as the heat dissipation layer in the IPL process. As a result, the sintered Cu films were obtained without heat damage to the polymer substrate (Figure 4c–h).

Plasma is a charged gas with strong Coulomb interactions. Utilizing its extremely active plasma, Suganuma’s group have developed the low-temperature plasma sintering on PVP-protected Cu particle pastes [56] (Figure 4i). A mixture of argon and hydrogen was used to generate active species during the plasma sintering process. In plasma sintering, the active species can clean the surface of Cu particles to achieve sintering at low temperatures. By the enhanced plasma effect, a Cu pattern with a resistivity of 15.9 μΩ·cm was obtained with only 10 min of plasma sintering at 200 W. No damage was achieved on the flexible polymer substrates due to a temperature below 75 °C during the plasma process (Figure 4j). A V-shaped Cu antenna pattern has also been fabricated on PET for a flexible antenna device application.

#### 4.1.2. Gelatin

Gelatin is widely used in the food industry as a key ingredient in foods, such as sweets, yogurt, nutritional supplements, and precooked deli dishes. It is composed of polypeptides having various functional groups, such as amino groups, mercaptans, and carboxylic acid groups in the side chains. These functional groups can effectively coordinate with the particle surface, and the gelatin surface layer causes Cu colloid stabilization and oxidation resistance of Cu particles under ambient conditions, as is the case of PVP polymers.

Yonezawa’s group has developed gelatin-protected Cu particles (c.a. 130 nm) for Cu-based pastes. An oxidation-reduction two-step sintering process was used for the Cu particles to obtain highly conductive Cu film at low-temperature sintering [57,58,59]. In the first step of Cu particle oxidation at the annealing in air, Cu_2_O prominences were generated on the Cu particle surface, which connect together through the organic layers. In the second step of the reductive annealing under Ar (96%)–H_2_ (4%) gas, these Cu_2_O prominences could be reduced to generate metallic Cu nanoparticles, which results in the necking and connection formation between the Cu particles (Figure 5a–c). The highly conductive Cu film (8.2 μΩ·cm) was obtained by the two-step sintering process at 200 °C, even though the gelatin surface layers of the Cu particle did not fully decompose at the sintering temperature [58]. Thus, Cu_2_O nano-projections play a significant role in the oxidation-reduction two-step sintering process at low temperatures. In their system, the polymer components in the Cu paste did not decompose at the sintering temperature; however, oxygen/hydrogen molecules can penetrate the softened gelatin layer above Tg to allow the oxidation-reduction in the Cu surface [59]. This indicates that the glass transition temperatures of the polymer components in the conductive pastes are also the critical temperatures considered for the low-temperature sintering of polymer-protected Cu nanoparticles.

#### 4.1.3. Polyethylene Glycol (PEG)

PEGs are an environment-friendly solvent with unique properties in terms of their cation complexation ability, being acyclic analogs of crown ethers [60]. PEGs act as both a size controller and a protective agent for the synthesis of Cu particles [61]. Zhang et al. reported the synthesis of Cu particles of 135 ± 30 nm prepared using Cu(OH)_2_ as the Cu precursor, L-ascorbic acid as the reducing agent, and PEG-2000 as the protective agent for applications in conductive inks [62]. The PEG-protected Cu nanoparticles could be stored for at least 1 month without any oxidization, whereas the prepared Cu nanoparticles without the PEG were easily oxidized. The PEG-protected Cu nanoparticle ink was screen-printed onto the PI substrate, and the conductive Cu patterns with a resistivity of 15.8 μΩ cm were achieved at a sintering temperature of 250 °C for 30 min under an N_2_ atmosphere (no reductive gas) (Figure 5d,e).

### 4.2. Surface Designs by Small Organic Ligands

Various protective small organic ligands, such as organic acid, alkylamine, alkanol amine, and alkanethiols, were used to hinder the oxidation of Cu nanoparticles in air. Compared with the case of polymers, such small organic ligands adsorbed on Cu nanoparticles thermally decompose at relatively lower temperatures, and reducing agents are often produced through thermal decomposition. In some cases, the reduction ability of small organic ligands suppresses Cu oxidation to facilitate conductive Cu films with low resistivities at low sintering temperatures. Herein, we focus on such small organic ligands for Cu-based inks/pastes. The surface designs by small organic ligands for Cu-based inks/pastes are summarized in Table 2.

#### 4.2.1. Oleylamine

Oleylamine (OAM) can be used for synthesizing stable Cu nanoparticles as the protective ligands [63,64]. It tightly binds to the surface of Cu nanoparticles via the Lewis bases of amines to donate its lone pair of electrons. Dai et al. reported OAM-protected Cu nanoparticles, which were synthesized by the thermal decomposition of Cu formate (CuF) using OAM as both the complexing ligand and stabilizing agent. The particle size of Cu nanoparticles could be controlled by the ratio of OAM/CuF = 2:1, 3:1, 4:1, and 5:1 to produce average diameters of 25.8, 14.7, 10.6, and 9.4 nm, respectively [65]. Although the particle surface was partially oxidized to Cu_2_O in air, Cu_2_O could be reduced to metallic Cu during thermal sintering due to the release of H_2_ through the Cu nanoparticle-catalyzed decomposition of OAM at a temperature higher than 150 °C (Figure 6a). The OAM-protected Cu nanoparticle (10.6 nm) inks provided the Cu film with a resistivity of 84.2 μΩ cm at the sintering temperature of 300 °C under an N_2_ atmosphere. Recently, Oliva-Puigdomènech et al. further developed the upscaled reaction to a 1 L reaction volume to produce over 50 g of OAM-protected Cu nanoparticles [66]. Notably, they paid attention to the oxidation suppression during Cu nanoparticle synthesis under ambient conditions. The oxidation of Cu nanoparticles depended on the concentration of the Cu precursor. The metallic Cu content of oxidized Cu nanoparticles increased from 8.8 to 97.3% with an increase in the concentration of the precursor from 2 to 40 g/L due to the reduction in the initially dissolved oxygen via the formation of Cu_2_O by the reaction of Cu with O_2_. In addition, faster heating rates (in the ranges of 2.5, 5, 10 and 20 °C/min) or higher reaction temperatures (in the ranges of 120, 140 and 160 °C) suppress the oxidation of Cu nanoparticles even under ambient conditions. The Cu nanoparticle-based ink provided Cu patterns with a resistivity of 23 μΩ cm at the sintering temperature at 400 °C for 1 h under an N_2_ atmosphere.

#### 4.2.2. Alkanolamine

Alkanolamines are bifunctional molecules having both hydroxyl (–OH) and amino (–NH_2_, –NHR, or –NR_2_) groups on an alkane backbone. As a result, they have a wide variety of properties that are common to amines and alcohols. Hokita et al. reported a high-concentration (up to 0.6 M Cu salt) synthesis of sub-10-nm Cu nanoparticles with sizes of 3.5 ± 1.0 nm in ethylene glycol under ambient condition using 1-amino-2-propanol (AmIP) as a protective agent [67]. The metalacyclic coordination stability of a five-membered ring type between the Cu and AmIP provided the superior durability of the AmIP-protected Cu nanoparticles in a solution. In addition, the secondary OH group of AmIP has the ability to reduce the Cu(I) to Cu(0) during the thermal sintering process of Cu. The AmIP-Cu nanoparticle inks produced Cu films with a resistivity of 30 μΩ cm after thermal heating at 150 °C for 15 min under an N_2_ flow. The low resistivity of the sintered Cu film (10^−5^ Ω cm) was also confirmed even after air exposure of around 4 months. Sugiyama et al. synthesized Cu nanoparticles protected by different types of alkanolamines: AmIP, 1-amino-2-propanol (AB), and 3-amino-1-propanol (NPA). The adhesion to a PI film was compared for films prepared from these inks. Only the Cu film prepared from the AB-protected Cu nanoparticle ink exhibited a strong adherence to the PI substrate without a decrease in the electrical conductivity (52 μΩ cm). The carboxylic compounds from the AB ligands, which were produced during the thermal sintering of the Cu inks, acted as adhesion promoters to strengthen the adhesiveness of the sintered Cu film to the PI substrate [68].

Li et al. developed a new type of Cu nanoaggregates coated with sub-10-nm AmIP-Cu nanoparticles for Cu-based inks. The Cu nanoaggregates have a size of more than 200 nm, but they have a relatively loose structure covered with the shell layer of small AmIP-Cu nanoparticles [69] (Figure 6b). The shell of nanoaggregates enhances the sintering properties at low temperatures. After sintering at 250 °C under an atmosphere of Ar-H_2_ (5% H_2_) gas mixture, the sintered Cu film exhibited a low resistivity of 4.37 μΩ·cm, which is only 2.5 times larger than that of Cu bulk (Figure 6c,d).

#### 4.2.3. Short-Chain Carboxylic Acids

Carboxylic acids, such as oleic acid and lauric acid, can act as protective agents by coordinating with the surface of Cu nanoparticles, thereby playing a significant role in preventing the Cu particles from undergoing oxidation. Short-chain carboxylic acids weakly binding to the Cu surface are favorable for an effective removal of the protective ligand from Cu nanoparticles during the thermal sintering process. The copper oxide removal activity of short-chain carboxylic acids is also attractive. Herein, we describe short-chain carboxylic acids, such as lactic acid, ascorbic acid, and citric acid for Cu nanoparticle inks.

Lactic acid: Deng et al. synthesized Cu nanoparticles protected by various short-chain carboxylic acids (acetic acid, glycolic acid, alanine, lactic acid, and citric acid) via the reduction in Cu acetate with hydrazine in ammonia-water solution [70]. When using lactic acid, stable Cu nanoparticles with sizes of less than 10 nm were obtained; they had the lowest surface oxide content. The advantages of lactic acid include the low decomposition temperature, antioxidation, and reductive capability as protective agents for Cu-based ink. The sintered Cu film from lactic acid-protected Cu nanoparticle ink exhibited a low electrical resistivity of 21.0 and 9.1 μΩ·cm after annealing at 150 and 200 °C for 60 min under an N_2_ atmosphere, respectively.

Ascorbic acid: Strong and toxic reducing agents, such as NaBH_4_ or hydrazine, are often employed for the synthesis of Cu nanoparticles. However, the use of such toxic reducing agents may not be preferred for the large-scale synthesis of Cu nanoparticles. Thus, cost-efficient and environment-friendly synthesis methods are desirable with no use of toxic agents. Ascorbic acid (AA), which is a mild/nontoxic agent, has been used for the synthesis of Cu nanoparticles as a reducing/protecting agent [71,72]. The reason for the excellent dispersion stability of Cu nanoparticles in water is the oxidation product of L-ascorbic acid (i.e., dehydroascorbic acid) adsorbed on Cu nanoparticles [71]. More importantly, the generated dehydrogenated AA on the surfaces of Cu nanoparticles can prevent the oxidation of Cu. Zhang et al. reported that the size control of Cu nanoparticles with mean particle sizes of 50, 150, 300, and 450 nm was achieved by regulating the dosage of the Cu source (Cu hydroxide) and reducing agent (AA) [73]. The as-prepared Cu nanoparticles exhibited a strong oxidation resistance, and they did not oxidize after being stored for 30 days under the atmospheric environment. The Cu nanoparticle-based ink was screen-printed onto a flexible substance. The resistivity of the conductive pattern was as low as 16.2 μΩ·cm upon heating at 240 °C (40 min) under an N_2_ atmosphere, where adjacent nanoparticles melted and formed clusters, which connected three-dimensionally, facilitating the formation of the conductive path (Figure 7a). A continuous square Cu circuit with a 0.5-mm conductor width was screen-printed on a flexible PI substrate (Figure 7b).

Citric acid: Citric acid (CA), which has a lower decomposition temperature (~200 °C) than AA, can act as a complexing and protective agent for the synthesis Cu nanoparticles [74]. Yokoyama et al. developed the synthesis of CA-protected Cu nano/microparticles by chemically reducing aqueous Cu–CA complexes using AA as a reducing agent [75]. The particle size depended on the pH of the solution during synthesis. The average size of the particles formed at pH 7.0 was 2 μm (Cu MPs), whereas that of the particles formed at pH 11.0 was 100 nm (Cu NPs). The resistivity of the sintered Cu films prepared using the Cu-based ink by heat treatment at 300 °C under Ar gas flow was improved from 470 to 8.2 μΩ·cm by mixing the Cu NPs with Cu MPs and washing Na ions on the particle surfaces with methanol. In metal NPs with a bimodal particle-size distribution, small NPs filled the voids between the neighboring large particles, which resulted in the generation of a well-packed Cu particle film. To further obtain the well-packed particulate structure, Yokoyama et al. utilized the capillary immersion force exerted during the drying of the inks using 1-propanol to cause the Cu NPs (particle size of 61.7 nm) to attract each other, thus resulting in tightly packed Cu nanoparticle films [76]. The resistivity (7.6 μΩ cm) of the sintered Cu films at 200 °C under Ar (98%) and H_2_ (2%) was much lower than that prepared by doctor blading using α-terpineol as a solvent.

#### 4.2.4. Oleic Acid

Oleic acid with a long chain length can be strongly chemisorbed onto the Cu surface as a protective molecule. Jeong et al. prepared air-stable, surface oxide-free Cu nanoparticles with a bimodal particle-size distribution of 42.3 and 108.3 nm for Cu-based ink [77]. The bimodal particle-size distribution of Cu nanoparticles leads to a well-packed particulate structure as well as a low-temperature densification reaction. In this synthesis, oleic acid, octylamine, and phenylhydrazine were used as a protective agent, solvent, and reducing agent, respectively. The oleic acid-protected Cu nanoparticles were stable in the air. XRD analysis is often employed to identify the presence of oxide phases (CuO, Cu_2_O) on synthesized Cu nanoparticles. However, the amorphous nature of the surface oxide and the small volume make it difficult to identify the presence of thin surface oxide. Thus, TEM and X-ray photoelectron spectroscopy (XPS) analyses offer more accurate information on the surface oxide layer. The former is used to visualize the nanoscale surface structure, whereas the latter is utilized for the identification of the oxidation states of Cu. The combined analysis of TEM and XPS supported the surface oxide-free Cu nanoparticles protected by the dense amorphous layer (~1 nm) of oleic acid [77]. The oleic acid chemisorbed to the Cu surface can be decomposed into a volatile moiety such as oleyl alcohol under a small amount of hydrogen at a relatively low temperature. The resistivity of sintered Cu film was measured to be 21, 3.9 and 3.5 μΩ·cm at the sintering temperatures of 200, 250 and 300 °C, respectively, under a mild hydrogen (10%) atmosphere.

However, even with surface oxide-free Cu nanoparticles, flexible plastic substrates cannot be used for oleic acid-protected Cu nanoparticle inks due to the high sintering temperature of above 250 °C. For the issue, they further employed IPL sintering for oleic acid-protected Cu nanoparticle inks to produce highly conductive Cu electrodes on various flexible substrates [78]. Resistivities of 6.7 μΩ cm and 51.2 μΩ cm were achieved for the IPL-sintered Cu electrodes on PI and PET, respectively. Additionally, the flexibility of the IPL-sintered Cu film electrodes was demonstrated without any significant variations in the resistivity, even after 10,000 cycle bending tests on three different substrates of PI, PEN, and PET. Subsequently, they employed Ag(lower)/Cu(upper) bilayer electrodes, and the Ag-based layer could suppress the propagation of thermal damage for underlying substrates. As a result, the Cu electrodes with a lower resistivity of 22.6 μΩ·cm were produced on large-area PET substrates using the IPL process [79].

Park et al. applied a transversally extended laser plasmonic welding process to oleic acid-protected Cu nanoparticles using a NIR laser at a wavelength of 1064 nm [80] (Figure 7c). The absorption length at the NIR region (at a wavelength of 1064 nm) caused a homogeneous densification of the Cu nanoparticle film. By optimizing the NIR laser power and the scan speed, a highly conductive Cu conductor with a resistivity of 4.6 μΩ·cm on a rigid glass substrate was produced, which was a lower resistivity than that (38.9 μΩ·cm) using a green laser (532 nm). Its applicability to cost-effective polymeric substrates, including PI, PEN, and PET, was tested with optimized processing conditions. The resistivity of Cu conductors on PI substrates was compatible with that on a glass substrate, whereas the Cu electrodes fabricated on a PET film exhibited a relatively high resistivity (85 μΩ·cm). Regardless of the kind of substrates used, including glass, PI, PEN, and PET, all the electrodes exhibited good adhesion properties.

#### 4.2.5. Alkanethiols

Alkanethiols bind organic ligands to the surface of Cu nanoparticles via the thiol moieties due to the strong Cu–thiolate bond. The Cu oxidation was inhibited by the protective layers formed by Cu–S bonding of 1-octanethiol for 35 days [81]. Her et al. prepared 1-octanethiol-protected Cu nanoparticles with an average diameter of 100 nm using the dry coating vaporized self-assembled multilayers (VSAMs) method to prevent the oxidation of Cu nanoparticles [82]. The thermogravimetry differential thermal analysis (TG/DTA ) curves indicated that 1-octanethiol could be removed at 150 °C, allowing the low-temperature sintering of the Cu nanoparticles. The Cu nanoparticle ink (10% Cu mass, a solvent of 1-octanol) provided a highly conductive film of 26.3 and 5.69 μΩ·cm sintered at 150 and 200 °C, respectively, under a hydrogen gas atmosphere. They further employed the IPL sintering method with preheating at 250 °C under hydrogen gas to inkjet patterns printed on a flexible PI film from the ink of Cu nanoparticles with a 1-octane thiol coating layer [83]. The optimal light-sintering condition was 24.7 J/cm^2^ of energy density from a 5 ms pulse, which resulted in a resistivity of 24 μΩ·cm. A high durability (1.45 times increase in resistivity after 1000 bending cycles at a radius of 2.5 mm) and long-term oxidation stability over two months of sintered Cu patterns were achieved, indicating its suitability for flexible electronic device applications.

### 4.3. Surface Designs by Core-Shell or Alloy Structure of Nanoparticle

The surface designs of the core-shell or alloy structure of nanoparticles for Cu-based inks/pastes are summarized in Table 3.

#### 4.3.1. Metal shell/alloy

The surface organic layers of the Cu surface effectively slow down the penetration of oxygen to the surface, thus decreasing the oxidation rate. The main disadvantage of this approach is the surface formation of the nonconductive organic layer of Cu nanoparticles. Another approach for obtaining stable Cu nanoparticles applicable for conductive ink is the formation of Cu-Ag core-shell (Cu@Ag) nanoparticles with a tunable Ag shell by the redox-transmetalation, where the oxidation of the Cu core is prevented by the Ag shell [84]. The Cu@Ag nanoparticles can be prepared via a galvanic displacement reaction when the surface of the preformed Cu core works as a reducing agent for the second metal (M) with a higher reduction potential. Herein, we mainly review Cu-metal core-shell (Cu@M), M = Ag, Ni, and Sn. Their alloy nanoparticles for conductive inks are also described.

Cu-Ag: Among the candidates as the second metal, Ag is the best material to prevent Cu from oxidation because of its high conductivity. Pajor-Świerzy et al. reported the synthesis of Cu@Ag particles with a Cu core (~1 μm) and a thin Ag shell (~20 nm) for printed electronics [85]. First, Cu particles were synthesized using Cu nitrate (metal source), poly acrylic acid (PAA) sodium salt with MW8000 (protective agent), Ag nanoparticles (catalytic amounts to accelerate the formation of Cu nanoparticles), and sodium formaldehyde sulfoxylate dihydrate (SFS, reducing agent). The SFS is less toxic and less dangerous, compared with conventional reducing agents (hydrazine or sodium borohydride). At the next stage, after washing out the excess reducing agent, the transmetalation reaction was performed by adding an ammonia complex to AgNO_3_ to dispersed purified Cu particles, resulting in the formation of air-stable Cu@Ag particles (Figure 8a).

The absorption spectrum of colloidal Cu@Ag nanoparticles and the XRD pattern showed the presence of both Cu and Ag in the nanoparticles (Figure 8b,c). The XRD analysis also demonstrated no oxide formation of Cu during 6 months of storage. The Cu@Ag nanoparticle ink (metal mass 30 wt%, propylene glycol as the solvent, and 0.05% of BYK 348 as a wetting agent) produced a highly conductive film with a resistivity of 11.34 μΩ·cm at a sintering temperature of 250 °C under an N_2_ atmosphere for 15 min.

Yu et al. synthesized PVP-protected Cu@Ag nanoparticles with an average particle diameter of 50 nm, including an Ag shell of thickness from around 2 to 10 nm [86]. The TGA analysis indicated that the oxidation of pure Cu nanoparticles occurred at around 85.7 °C under ambient conditions. Contrarily, Cu@Ag nanoparticles were oxidized at a higher temperature of around 156.3 °C due to the dewetting of the Ag shell at that temperature, which led to the exposure of the Cu core to air (Figure 8d). The XPS analyses revealed that the surface of Cu@Ag nanoparticles was negligibly oxidized even after 3 weeks, which could be attributed to the protection of the Ag shell on the surface. The Cu@Ag nanoparticle ink produced the electrical resistivity of 27 μΩ·cm at a sintering temperature of 350 °C under a reductive atmosphere (5% H_2_) for 60 min (Figure 8e). A sintering mechanism for Cu@Ag was proposed, in which Cu-Ag core-shell nanoparticles have an enhanced sintering performance owing to the dewetting and necking of the Ag shell (Figure 8d).

Tan et al. synthesized cetyl trimethyl ammonium bromide (CTAB)-protected Cu@Ag nanoparticles with an average particle diameter of 15 nm [87]. Such small-sized Cu nanoparticles are easily oxidized into either Cu_2_O under ambient conditions, but the core-shell structure offers the oxidation resistance of nanoparticles. Furthermore, the Cu@Ag conductive pattern was directly drawn on ordinary photo paper using a roller pen filled with 30 wt% Cu@Ag nanoparticle ink. The electrical resistivity of the pattern was as low as 13.8 μΩ cm at a sintering of 150 °C for 1 h under an N_2_ atmosphere.

Li et al. prepared a mixture of Cu@Ag particles (1.7 μm in size) and Ag nanoparticles (60 nm in size) [88]. The mixed inks with different sizes improved the particle packing density for the highly conductive flexible patterns. The mixed inks generated a pattern with a low resistivity of 5 μΩ cm at a low temperature of 160 °C under an argon atmosphere for 2 h. Even after bending over 3000 cycles and storage for 300 days, the resistivity of the patterns exhibited a small increase from 5 to 20 μΩ cm, which was about two orders of magnitude lower than that of used single Cu and Ag particles. This performance is acceptable for conductive patterns in practical applications.

Zhang et al. synthesized elliptic Cu-Ag alloy nanoflakes with a high purity and uniformity present size of 700 × 500 nm and a thickness of 30 nm via in situ galvanic replacement between prepared Cu particles and Ag ions [89]. PVP polymers promoted the anisotropic growth in the nanoflake as well as the dispersion stability. The Cu-Ag alloy nanoflakes were utilized as fillers for conductive paste in an epoxy resin matrix. Conductive patterns on flexible PET/PI substrates had a resistivity of 37.5 μΩ·cm after annealing at 150 °C for 2 h. Compared with the traditional Ag microflakes (52.6 μΩ·cm), the alloy nanoflakes provided an improved conductive interconnection due to their nanoscale thickness. Moreover, the conductive patterns on PET or PI substrates retained good conductivity even after hundreds of repeated bends at different angles. Therefore, alloy nanoflakes could be a promising conductive filler candidate for flexible printed electronics and other conductive applications.

Dou et al. reported the synthesis of nearly monodisperse Cu-Ag alloy nanoparticles with Ag and a Cu molar ratio from 1:9 to 9:1 using a thermal decomposition method (Ag acetate and Cu acetate as the metal source, 1-octadecanol as a nontoxic reducing agent, oleic acid and oleylamine (OAM) as protective agents, and 1-octadecene as a solvent) [90]. The alloyed Cu1Ag1 nanoparticles (Cu/Ag = 1:1) had a size uniformity of 10.95 nm with relative standard deviation (RSD) = 4.89%, and they exhibited improved properties against oxidation. The Cu1Ag1 nanoparticle inks, including ethylcellulose, terpineol, hexane, and polymeric matrix, exhibited a resistivity of 58 μΩ·cm after annealing at 250 °C under a H_2_ and N_2_ (5:95 in volume) atmosphere for 3 h. Moreover, the electrical migration could be significantly reduced by adding some Cu to the Ag nanoparticles, and Cu7Ag3 nanoparticles have significantly suppressed the change in resistance after applying 1 V for 25 h. Two possible mechanisms were considered: (i) the Cu segregation at Ag grain boundaries retard Ag diffusion, and (ii) lower surface energy and fewer thermal grain boundary grooves in Ag/Cu grains compared with only Ag grains.

Cu-Ni: An alloying of Cu with Ni or a coating of Ni on Cu can be utilized to improve the oxidation resistance of Cu by forming a Ni-containing thin oxide layer. Kawamura et al. utilized a wire explosion process (Figure 9a) to produce Cu alloy nanoparticles in an aqueous solution of 200 mM AA, including 99Cu-1Sn, 95Cu-5Ag, 95Cu-5Ni, and 70Cu-30Ni [91]. The conductive films of as-prepared Cu nanoparticles exhibited electrical resistivities of 172 μΩ·cm by sintering at 200 °C in H_2_. Cu nanoparticles alloyed with 5% Ni and 30% Ni had electrical conductivities of 599 and 857 μΩ·cm, respectively. The resistivities of sintered Cu films dramatically increased from 172 μΩ·cm to 1.02 Ω cm after 24 h at 85 °C and 85% relative humidity(RH). Contrarily, the sintered 70Cu-30Ni film exhibited a relatively small decrease in resistivities from 857 to 7670 μΩ·cm after the corrosion test (Figure 9b).

Gu et al. developed a simple chemical process for inhibiting the oxidation of Cu-Ni (Cu55Ni45, constantan) alloy nanoparticles with a size of around 100 nm for the printed conductive patterns [92]. The high melting temperature of constantan (1221–1300 °C) makes the sintering process difficult, especially at low temperatures. The sintering process of constantan is usually conducted under a reducing atmosphere, including a flammable gas such as hydrogen and methane. Gu et al. used ammonia chloride (NH_4_Cl) to etch the surface oxidation (CuO and NiO), and the constantan nanoparticles can be sintered together using a thiourea dioxide (TD) additive which reduces the sintering temperature. The TD can reduce Ni^2+^ and Cu^2+^, which are produced by NH_4_Cl etching. With the combination of NH_4_Cl and TD treatment, the sintered constantan film at 350 °C for 2 h in vacuum exhibited a resistivity of around 400 μΩ·cm, and the temperature coefficient was the same as the bulk constantan.

Fang et al. reported Cu@Ni core-shell nanoparticles with an enhanced oxidation resistance. They synthesized the Cu@Ni core-shell nanoparticles via an injection using Cu seeds to induce the epitaxial growth of Ni shells [93]. The Ni shell-thickness could be controlled by varying the Cu/Ni molar ratios in the precursor solution. The injection rate of the Cu precursor changed the size of the Cu seeds, allowing the size control of the core-shell nanoparticles. The oxidation resistance of Cu@Ni films increased with an increase in the Ni/Cu ratio, whereas the resistivity increased with the increase in the Ni/Cu ratio. The low resistivity of 27 μΩ cm was achieved for sintered Cu@Ni (Cu/Ni = 8:1) films at 350 °C.

The Ni shell coating provides an advantage in terms of the oxidation stability of Cu nanoparticles; however, intrinsically, the presence of the Ni shell layer results in an increase in the resistance and sintering temperature, which are not desirable for flexible substrates of low heat resistance. To solve these problems, Kim et al. utilized an IPL sintering method to the ink of Cu@Ni core-shell nanoparticles with oxidation-resistant and highly conductive characteristics without the degradation of their electrical conductivity [94] (Figure 9c). The Cu@Ni nanoparticles were synthesized by adding the Cu nanoparticle solution to a flask containing a solution of Ni acetylacetonate, oleic acid, and phenylhydrazine in the OAM solvent. The mixture was purged with N_2_ for 60 min and then heated to 240 °C for an additional 60 min. The Ni shell thickness of Cu@Ni NPs was tunable by the Cu/Ni molar ratios in the precursor solution. The combination of continuous spray coating and IPL sintering produced large-area Cu-Ni conductors on various flexible substrates, such as PI, PEN, poly(ether sulfone) (PES), and paper substrates. The Cu@Ni (Cu/Ni = 5:1) electrodes showed a low resistivity of 52 μΩ cm at the optical energy dose of 1.59 J cm^−2^. Moreover, the Cu@Ni (Cu/Ni = 2:1) electrodes exhibited stable sheet resistances (ΔR/R_0_ < 1) even after 30 days of aging at 85 °C and an 85% relative humidity (Figure 9d). The light-emitting diode (LED) lamp exhibited no considerable deterioration after the humidity test (Figure 9d). Further, the electrode was mechanically stable against bending cycles with a bending radius of 10, 8, 6, and 4 mm, and the ΔR/R_0_ increased slightly (up to ~1) even after 1000 bending cycles.

Cu-Sn: Oh et al. synthesized Cu-Cu10Sn3 core-shell nanoparticles for producing highly conductive electrodes in combination with a large-area processable, continuous photonic sintering process, which allowed for the use of an R2R process [95]. The melting point (798 °C) of Cu10Sn3 phases is lower than that of bulk Cu (1059 °C). The Cu10Sn3 shell layer with a low melting point produced the highly conductive electrodes with resistivities of 27.8 and 12.2 μΩ cm under low energy dose conditions of 0.97 and 1.1 J/cm^2^, respectively. Park et al. prepared printable mixed inks comprising multicomponent ingredients of Cu, Ni, and Cu-Cu10Sn3 core-shell nanoparticles [96]. The sintered electrodes on the PI substrate exhibited a resistivity of 49 μΩ cm and excellent oxidation resistance with a normalized resistance variation of around 1 during the 30 days of 85/85 tests.

#### 4.3.2. Carbon Shell

Cu nanoparticles protected by C shells (Cu@C) have also attracted considerable interest, since the C shells can protect the Cu core from oxidation. Besides, the Cu@C nanoparticles exhibit an excellent dispersion stability in various polar solutions owing to the presence of numerous functional groups (–COOH, –CO, –OH, etc.) on the C surface without the need for additional dispersion surfactants.

Kim et al. synthesized Cu nanoparticles protected by thin defective C shells (Cu core size of 20 nm, C shell thickness of 1.0 nm) [97] (Figure 10a). The Cu@C nanoparticles were prepared via an electrical explosion of wires, a physical vapor condensation method applicable to the mass production of metallic nanoparticles with high purity. The Cu@C nanoparticles have no Cu oxides in the XRD spectra upon exposure to air at room temperature for 90 days, contrary to the faster oxidation of unprotected Cu nanoparticles. The outer C shells contained numerous defects, such as disorder, lattice distortions, and noncrystalline regions, as demonstrated in the D- and G-bands of the Raman bands (Figure 10b). The defective C layer can be removed from Cu@C nanoparticles at a C shell burn-off temperature of 180 °C. The Cu@C nanoparticles exhibited a thermal oxidation sequence of C → CO_2_, Cu → Cu_2_O, and Cu_2_O → CuO with a C burn-off temperature of 180 °C. The Cu@C-based inks produced sintered Cu films with 25.1 μΩ cm at two-step oxidation-reduction annealing process (i.e., oxidation for the thermal removal of C shells in the air and subsequent reduction in the Cu surface oxides at 200 °C under a 10% H_2_ atmosphere) (Figure 10c,d).

Cu nanoparticles protected by multilayer graphene-encapsulated (MLG-Cu) NPs can protect the Cu cores from oxidation and simultaneously serve as a conductive contact between individual MLG-Cu NPs without the sintering process [98]. Tseng et al. reported a one-step synthesis method to fabricate MLG-Cu NPs with sizes of 52 ± 9 nm directly on various substrates through an expeditious chemical vapor deposition (CVD) [99]. The electrical resistivity of the pristine MLG-Cu NPs on PI without thermal sintering was measured as 170 μΩ cm, which is ~100-fold lower than in earlier reports. The MLG-Cu NPs retained almost all their conductivities even after ambient annealing at 150 °C due to the MLG shell. Furthermore, the MLG-Cu NPs on PI exhibited an excellent mechanical durability (only 7% increase) after 1000 bending cycles with a radius of 1.7 mm. They also demonstrated that the MLG-Cu NPs on PI could be used as p source-drain electrodes in fabricating flexible graphene-based field-effect transistor devices.

#### 4.3.3. Copper Oxide Shell

The protective agents or metal shells (e.g., Ag, Au, Ni and Sn) have been explored by encapsulating Cu particles to inhibit Cu oxidation before and during printing, as described above. As other approaches, Cu oxide (or Cu@Cu oxide) nanoparticles are dispersed in reducing solvents to solve the oxidation problem when coupled with IPL (or laser) sintering, or a two-step oxidation-reduction annealing process. The surface designs by Cu oxide for Cu-based inks/pastes are summarized in Table 4. Cu oxide nanoparticles are more stable than their Cu nanoparticle counterparts. The IPL sintering can remove the oxide shells of Cu nanoparticles using the IPL-reactive protective agents, such as PVP, leaving a conductive Cu film in a short period of time (a few ms) under ambient conditions [100,101,102,103,104,105,106,107]. Ryu et al. explained this phenomenon by oxide reduction either via an intermediate acid created by light irradiation or by hydroxyl (–OH) end groups, which act like long-chain alcohol reductants [100] (Figure 11).

Paper has a high porosity, high permeability, and surface roughness, which limits the formation of continuous Cu films on paper, in contrast with smooth plastic substrates (e.g., PET and PI). Ohlund et al. fabricated the conductive patterns on the CaCO_3_ precoating paper deposited by inkjet printing from water-based CuO nanoparticle (approximately 10−300 nm) ink via IPL sintering (within 5 ms, using a single pulse of light) [101]. The CaCO_3_ precoating increased the viable range of exposure energy and improved the processing reliability. Additionally, the porosity of the CaCO_3_ precoating layer enabled the effective removal of the water and ink additives, thus reducing the risk of film damage and delamination during the IPL exposure. As a result, a low resistivity of 3.1 μΩ·cm was achieved for a 400-μm wide conductor. Albrecht et al. investigated the influence of various substrates (40 porous papers and polymer-based substrates), together with the IPL sintering parameters on the electrical properties of CuO nanoparticle inks [102].

Rager et al. developed the scalable and low-thermal-budget photonic fabrication of Cu patterns with resistivity values as low as 10 μΩ cm using an R2R-compatible pulse-thermal processing technique, which involves the reduction and subsequent sintering of ink-jet-printed CuO patterns onto flexible polymer substrates [103]. The introduction of an initial drying step of CuO nanoparticle ink under an ambient condition, after printing and before sintering, significantly improved the mechanical integrity and electrical performance of the Cu patterns.

Oh et al. reported the systematic investigation on the effect of Cu oxide shell thickness on the IPL sintering of PVP-protected Cu nanoparticle ink [104]. They prepared the Cu nanoparticles with different thicknesses of the Cu oxide shell, from 2.1 to 14 nm, by increasing the oxidation temperatures from 100 to 500 °C in the air. The oxidation state (Cu_x_O) depended on the oxidation temperature: a Cu_2_O layer on the surface of the Cu NPs below 100 °C, a complex oxide of Cu_3_O_2_ formed at about 150 °C, and a thin layer of CuO formed on the outer layer at temperatures above 200 °C. The PVP on the Cu nanoparticles can reduce Cu oxide shells in the IPL sintering. However, thicker layers of PVP and Cu oxide shells increased the resistivities of sintered Cu film due to the residual layers of PVP and Cu oxide on the Cu nanoparticles. Therefore, it was concluded that Cu nanoparticles with oxide shells less than 4 nm and an optimized PVP amount are essential for highly conductive Cu films in the IPL sintering process (Figure 12). It was also found that multipulsed flashlight irradiations could decompose thick PVP layers and reduce thicker Cu oxide shells effectively without damaging the Cu films.

Ryu et al. reported a two-step flashlight sintering process to reduce the warping of polymer substrates during the sintering of PVP-protected Cu nanoparticles (particle diameter: 20–50 nm) with the Cu oxide shells (>2 nm) [105]. The two-step sintering process consists of 12 J/cm^2^ preheating followed by 7 J/cm^2^ primary sintering, thus resulting in highly conductive Cu films with a resistivity of 3.81 μΩ·cm on the PI substrate without residual warping. Besides, the adhesion strength of the two-step sintered Cu ink film indicated a 5B level compared with that of a one-step sintered case of a 3B level. Chung et al. used a mixed ink of Cu nanoparticles (~100 nm) with oxide shells of 20 nm and CuO nanoparticles (50 nm) for multiple IPL sintering to increase the light absorption efficiency and further decrease voids between Cu nanoparticles in the Cu film [106]. With an optimal Cu/CuO weight ratio of 1:80, the resistivity of the Cu film reduced as low as 6.5 μΩ·cm on PET, with the single energy at 3.08 J/cm^2^. This ink could be printed on PET films as conductive tracks with a high adhesion for the flexible electronics application.

Kwon et al. utilized L-ascorbic acid-protected Cu_2_O nanoparticles, which were prepared from the chemical reduction in Cu ions with L-ascorbic acid, for IPL sintering to design Cu patterns on a flexible substrate [107]. They compared the electrical resistivity of sintered Cu patterns from Cu_2_O or Cu nanoparticles. A superior electrical resistivity of 4.2 μΩ·cm from the Cu_2_O nanoparticle ink was obtained, which was lower than that of Cu nanoparticles (59 μΩ·cm). The Cu_2_O nanoparticles have a high UV absorbance and short visible regions, and these characteristics are most suitable to use the light energy for IPL sintering. In addition, the L-ascorbic acid absorbed on the Cu_2_O nanoparticles allows for the self-reduction of Cu_2_O nanoparticles to pure metallic Cu.

Yong et al. reported a novel approach to obtain highly conductive Cu films from CuO particles via the oxidation-reduction steps [108]. First, metallic Cu particles (~280 nm) were prepared from CuO particles by a chemical reduction with ascorbic acid. Second, the oxidative preheating process of the Cu particles at 200 °C in air produced the Cu oxide convex surfaces of nanoparticles. These Cu oxide convex surfaces facilitated the coalescence of particles with tight connections. After the reductive sintering process under the condition of 3% hydrogen-containing N_2_ gas at various temperatures of 200, 250 and 300 °C, the oxidative films were reduced entirely to metallic Cu films with low resistivities of 12.2, 7.8 and 5.6 μΩ·cm, respectively. The role of oxidative preheating is not only to remove the organic layers but also to build close connections between particles and form highly compact films by the convex oxide surfaces.

## 5. Surface Activation of Cu Micro/Nanoparticles for Low (or Room)-Temperature Sintering

Smaller nanoparticle sizes can reduce the melting temperature (T_m_) of Cu nanoparticles [109]. Thus, decreasing the size of Cu nanoparticles is an effective approach for low-temperature sintering, since the sintering temperature is 2/3–3/4 T_m_ [110]. Another strategy for low-temperature sintering is the surface activation of Cu particles. Herein, the term “surface activation” means to trigger sintering such as chemical reduction-assisted sintering, the removal of the surface oxide layer, and active species formation of the surface to facilitate the sintering. In this section, we describe such a surface activation approach of Cu-based inks. The surface designs by the surface activation for Cu-based inks/pastes are summarized in Table 5.

### 5.1. Surface Activation of Micron-Sized Cu Particles for Low-Temperature Sintering

Small Cu nanoparticles (in particular, sub-10-nm Cu nanoparticles) have a disadvantage in terms of their instability (i.e., oxidation and aggregation) and they require a large amount of protective agent for stabilization. In contrast, micron-sized Cu particles have an advantage on the stability and small amount of protective agent. However, micron-sized Cu particle powders have higher sintering temperatures compared with small Cu nanoparticles. Therefore, micron-sized Cu particles are not preferably used singly for low-temperature sintering. Wu et al. achieved a low-temperature sintering of micron-sized Cu powders (Cu flakes) employing the surface activation approach with ethanol vapor reduction [111]. The resistivity of the sintered Cu films is about 1000 and 100 μΩ·cm even at 120 and 140 °C under ethanol vapor, respectively. At the same annealing temperature, the electrical resistivity of sintered Cu film from spheroidal Cu particles was much higher than that of Cu flakes. The Cu flake has more contact area due to the face-to-face arrangement, which helps in the diffusion of Cu atoms associated with the sintering process. The mechanism of this low-temperature sintering is suggested to be the reduction in native oxide on Cu flakes by the ethanol vapor. The fresh surface of reduced Cu flakes is very active and tends to sinter via the face-to-face contact to lower the total surface energy.

Qi et al. reported the rapid low-temperature sintering in the air of large Cu particles (size ranges from 400 to 1200 nm) with synergistic surface activation and antioxidative protection [112]. The Cu-based paste was prepared by mixing the Cu particles with formic acid and 3-dimethylamino-1,2-propanediol (DMAPD). The Cu oxide on the surface of Cu particles was converted to Cu formate by formic acid. The DMAPD could promote the low-temperature decomposition of Cu formate by forming a Cu-amine complex (Cu–HCOOH–DMAPD) as the surface-active layer. In addition, the surface-active layer of Cu–HCOOH–DMAPD could suppress the oxidation of Cu particles in the air. The Cu–HCOOH–DMAPD complex could decompose at a lower temperature to produce fresh Cu nanoparticles on the surface of micron-sized Cu particles, thus resulting in the easy fusion and binding to the surrounding particles (Figure 13a). Thus, the mixed solvent of formic acid and DMAPD can activate the surface of micron-sized Cu particles by forming the Cu–amine complex. As a result, the resistivity of 63 μΩ cm was achieved after sintering at 200 °C in the air for 10 s (Figure 13b).

### 5.2. Surface Activation of Cu Micro/Nanoparticles for Room Temperature Sintering

As for Ag nanoparticle inks, several researchers reported room temperature sintering without heating or vacuum [113,114,115,116,117]. The detachment of protective agents from the nanoparticles’ surfaces by dipping solvents [113] or the use of destabilizing agents, such as Cl^−^ ions [114,115,116] and small tris (2-aminoethyl) amine [117], enables their coalescence and sintering at room temperatures. The room temperature sintering is also called “chemical sintering” or “self-sintering.” In the case of Cu-based inks, conversely, there are two critical factors in room temperature sintering: (i) the removal of the surface oxide layer and (ii) the detachment of protective agents from Cu nanoparticles. The former is essential for Cu-based inks in contrast to Ag-based ink since the presence of Cu oxide on the surface prevents Cu nanoparticle sintering.

Wu et al. reported the sintering of Cu particles at room temperature using a reduction-assisted method with ascorbic acid (AA) [118]. Micron-sized Cu powder (0.05 g) was immersed in an aqueous solution of 0.5 M AA (5 mL) for 6 h. The mixture was deposited on flexible polymer substrates and vacuum-dried at room temperature, producing a Cu conductive pattern with a resistivity of around 3000–8000 μΩ cm on various polymer substrates. With this process, the Cu conductive lines were fabricated on various substrates of PC, PI, and poly(methyl methacrylate) (PMMA), PEN, and PET films. The room temperature sintering mechanism was proposed as follows: the native oxide layers on the surfaces of Cu particles prevent the sintering, but they are eliminated by AA reduction. The removal of the surface oxide layer induces the rearrangement of nearby fresh Cu atoms which are very active, resulting in the self-sintering to merge with each other to minimize the surface free energy. The AA could act both as a reducing and adhesive agent, and thus, the excellent adhesion of Cu patterns after treatment with AA was demonstrated.

Lee et al. reported room temperature sintering by employing sub-10-nm Cu nanoparticles treated with poly(vinyl imidazole-co-vinyl trimethoxy silane) and 1-amino-2-propanol (AmIP) [119]. The sub-10 nm Cu nanoparticles enabled the formation of integrated structures even without heat treatment. The Cu nanoparticle paste was printed on a glass substrate and dried under ambient conditions for 3 to 4 h to allow the solvent to evaporate. The Cu conductive pattern exhibited a resistivity of 12,000 μΩ·cm without heat treatment. The Cu nanoparticle paste has also been successfully applied to make a dipole tag antenna, demonstrating the potential of sinter-free Cu NP paste for versatile applications.

Dai et al. developed Cu@Ag nanoparticle ink for room temperature sintering [120]. First, they synthesized oleyamine (OAM)-protected Cu@Ag nanoparticles with an average diameter of 11.7 nm. They conducted the room temperature sintering using a two-step process, which consists of (i) ligand exchange from OAM to AmIP and (ii) an immersion in an NaBH_4_ solution as the destabilizing/reducing agent (chemical sintering). After the two-step process in the air, the resistivity of the resultant metal pattern reached 36.3 μΩ·cm and exhibited proper adhesion (ASTM 5B) to the PET substrates. More recently, Dai et al. achieved a highly conductive Cu film with a resistivity of less than 20 μΩ·cm at room temperature sintering in the air via a chemical sintering mechanism [121]. In this process, OAM adsorbed on Cu particles with an average diameter of 11.9 nm was effectively eliminated via the reactive desorption by a methanol immersion, including 10 vol% folic acid, and at this stage, Cu^2+^ ions were generated on the surface. Next, the oxide (Cu_2_O) and Cu^2+^ ions on the Cu nanoparticle surface could be reduced to Cu by a 0.75 wt% NaBH_4_ solution immersion for 9 min (Figure 13c). The resultant Cu film exhibited a low resistivity of 16.93 μΩ·cm and also an excellent adhesion onto the PET film (Figure 13d).

## 6. Formulation Designs in Cu-Based Mixed Inks/Pastes

Cu-based mixed inks make up more than two conductive materials. The mixing of the conductive materials has been performed for the following purposes: (i) increasing the particle packing density, less small pores, and fewer cracks, (ii) low-temperature sintering, and (iii) increasing the mechanical reliability of sintered Cu films. Herein, we introduce formulation designs in Cu-based mixed inks: (i) Cu nano/microparticles, (ii) MOD/Cu particles, (iii) Cu nanowires/Cu nanoparticles (or MOD inks), and (iv) carbon nanotubes (CNTs)/Cu nanoparticles (or MOD ink). The formulation designs in Cu-based mixed inks/pastes are summarized in Table 6.

### 6.1. Cu Nano/Microparticles

Cu nanoparticle inks often suffer from the formation of cracks and pores in sintered Cu films due to the large volume shrinkage that deteriorates the electrical conductivity of the printed circuits [122]. The influences of the ink solvent, the solid fraction of the ink, the drying treatment, and the sintering parameters affect the crack formation. Dai et al. reported a mixed paste of Cu micro- and nanoparticles to prevent cracking and achieved an improved packing density [123]. The particle mixture of two different sizes reduced the porosity of the micro-paste and resolved the cracking issue in the nano-paste. The in situ temperature and resistance measurements indicated that the mixed paste had a lower densification temperature. The mixing of Cu nano/microparticles also revealed a ~12 times lower sheet resistivity of 0.27 Ω/sq and a ~50% lower porosity. The 3:1 (micro/nano, wt%) mixed paste was found to have the most substantial synergistic effect. Tam et al. also prepared mixed paste with Cu nanoparticles (60.8 nm) and Cu flakes (9.3 μm) [124]. The optimal formulation of mix pastes for the screen printing was determined to be 20 wt% Cu flakes and 80 wt% Cu nanoparticles, leading to a resistivity of 29 μΩ·cm for the sintered Cu film on PET at 120 °C under a 5% H_2_ atmosphere for 3 h, which was a lower resistivity (59 μΩ·cm) than the sintered Cu film from the paste with only Cu nanoparticles without Cu flakes. Kanzaki et al. fabricated conductive Cu films on PI and PET by low-temperature sintering under an air atmosphere using a mixed Cu paste of Cu particles (~300 nm) and AmIP-protected Cu nanoparticles (3–5 nm) with a 3:1 wt% mixture [125]. The oxalic acid was also added to prevent Cu oxidation during sintering in the air. The sintered Cu films exhibited a minimum resistivity of 55 μΩ·cm when sintered in the air at 150 °C for 10 s. Under an N_2_ atmosphere, a low resistivity of 8.4 μΩ cm was achieved for the sintered Cu film at 120 °C (Figure 14).

Several groups utilized mixed Cu pastes with different sizes for the IPL sintering process in the air [126,127,128,129,130]. In the IPL sintering process, Cu nanoparticle inks often face two problems: (i) the crack formation from the volume shrinkage under tensile stress on a flat substrate and (ii) the thickness limitation of fewer than 5 μm due to the high absorption efficiency of Cu nanoparticles. To solve these problems, Park et al. developed a printable paste from a mixture of micron-sized Cu flakes and oleic acid-protected Cu particles (Cu NPs) [126]. In the mixed paste, Cu flakes play a dominant role in allowing the extended penetration of white light through particulate layers and preventing the generation of a crack in the IPL sintering process. By adjusting the optical behaviors of the particulate layers by varying the composition of the mixture, a crack-free, thick Cu film with a thickness of 13.2 μm could be generated with a resistivity of 11.4 μΩ cm from the mixed pastes (Cu flake/Cu NP = 6:4) through the IPL sintering process in the air (Figure 15a–c).

Ryu et al. demonstrated the optimal IPL-sintered Cu nano/microparticle (CuNP/MP) film using the vacuum pressure for substrate holding, 150 °C substrate heating, and irradiation energy, 3.5 J/cm^2^ (Figure 15d) [127]. The IPL-sintered Cu film had a low resistivity of 6.94 μΩ·cm and 5B level of adhesion strength with almost no warpage of the PI substrate (Figure 15e,f). Additionally, the substrate heating by heating wires reduced the temperature difference in the Cu layer and PI film, allowing a fully dense structure of the IPL-sintered Cu structure and slight substrate warpage. Ryu et al. further investigated the effect of the rheological properties of Cu nano/microparticle paste with various amounts of dispersant on the printability and flashlight sinter-ability [128]. More recently, they reported the effect of UV surface modifications on a PI substrate for the IPL sintering of Cu nano/microparticle paste [129]. The IPL-sintered Cu film from the mixed paste provided no substrate damage despite the high irradiation energy since the UV-generated hydrophilic groups on the PI substrate facilitated thermal conduction and helped strengthen adhesion. As a result, the optimized IPL-sintered Cu nano/microparticle film on the UV-modified PI substrate exhibited a low resistivity of 5.94 μΩ cm and a high adhesion strength level of 5B.

Unlike Ag-based ink, Cu-based ink usually needs very high energy photonic sintering to convert Cu oxide into metallic Cu, which often causes electrode destruction due to an unreleased stress concentration and massive heat generation. Chen et al. developed a Cu-Sn mixed ink by mixing Cu (150 ± 30 nm in diameter, oxide thickness <3 nm) and Sn particles (100–300 nm in diameter) [130]. The sintered energy of the Cu/Sn film with a mass ratio of 2:1 (Cu/Sn) was found to be 21% less than that of only a Cu film, which is attributed to the lower melting point of Sn for mixed ink. In the IPL sintering process, the Sn particles were effectively fused among Cu particles and formed a conducting path between them. The Cu–Sn mixed ink could make metal-mesh transparent conductive films (TCFs) with a line width of 3.5 μm, high transmittance of 84%, and low sheet resistivity of 14 Ω/sq. The great benefit of lowering photonic sintering energy is the reduced defect level in the TCFs in the IPL sintering process.

### 6.2. Metal-Organic Decomposition (MOD)/Cu Particles

Metal-organic decomposition (MOD) ink consists of metal precursors dissolved in volatile solvents or organic complexing agents that act as solvents. Cu oxidation and agglomeration do not occur in MOD ink because the metal exists in an ionic form in the inks. Besides, the ink formulation process is straightforward only by mixing a metal precursor, complex agent, and solvent, in contrast to the particle-based ink with synthesis, washing, and stabilizing (Figure 16). In most cases, Cu formate (CuF) is selected as the metal precursor in the Cu MOD ink owing to its self-reducible properties. Additionally, Cu formate can be complexed with alkylamine or alkanol amines, such as 2-amino-2-methyl-1-propanol (AMP), resulting in a Cu formate–amine complex. The CuF–amine complex can be directly transformed to a conductive Cu film via a low-temperature thermal decomposition process, typically involving an in situ nucleation and the growth of Cu nanoparticles. Recently, several excellent reviews have discussed MOD inks [131,132], and readers can refer to the detail of MOD inks. However, there is still a significant problem with MOD inks. The existence of excess organics and volatile gases (e.g., H_2_, H_2_O and CO_2_) causes undesired voids and pores in the Cu films. The low Cu load (typically less than 20 wt%) makes the production of a thick Cu film challenging. To solve the issues in the MOD inks, Cu nano/microparticles are mixed with the MOD inks to increase the Cu load and the decrease the production of volatile gases. Herein, we discuss the mixed inks of MOD and Cu particles.

Yonezawa‘s group added Cu particles with an average size of 0.8 μm to a Cu MOD ink composed of Cu(II) formate and 1-amino-2-propanol (AmIP) to increase the metal load to 65 wt%. They obtained a uniform conductive pattern with a resistivity of 900 μΩ cm at a low sintering temperature of 100 °C under an N_2_ atmosphere [133]. They further examined the effect of the Cu particle sizes (0.4, 0.5, 0.8 and 2.5 μm) in the mixed pastes on the resistivity of sintered Cu films [134]. The lowest resistivity of 32 μΩ cm was achieved using 0.4-μm Cu particles at a sintering temperature of 120 °C for 30 min under an N_2_ atmosphere. They also prepared Cu particles (154 ± 54 nm in size) protected with an amine-decomposable polymer (poly(propylenecarbonate), PPC) and mixed the PPC-protected Cu particles and CuF–AmIP complex to achieve a low resistivity at low-temperature sintering under an N_2_ atmosphere [135]. PPC can be decomposed into smaller molecules via an aminolysis reaction with AmIP. The smaller-size and wide-size distributions were beneficial for increasing the packing density of the sintered Cu films due to less vacancy between particles after sintering. The synergistic effects arising from the amine-decomposable Cu particles and the CuF–AmIP complex offered the sintered Cu film with a resistivity of 88 μΩ cm at 100 °C for 1 h under an N_2_ atmosphere.

Suganuma’s group developed mixed pastes consisting of Cu particles with an average size of 0.7 μm and a CuF–AMP complex [136]. The sintered Cu pattern from the mixed paste achieved a low resistivity of 11.3 μΩ cm at 140 °C for 15 min under an N_2_ atmosphere. Contrarily, the use of small Cu particles of 0.17-μm size in the mixed paste achieved a lower conductivity compared with the case of the 0.7-μm size due to the easy oxidation of smaller Cu particles and numerous interparticle contact resistances. The fresh Cu nuclei from the thermal decomposition of the CuF–AMP complex homogeneously attach to the Cu particle surface via the heterogeneous nucleation process, resulting in the connection and neck growth between the Cu particles to achieve highly conductive Cu patterns (Figure 17a,b). Furthermore, the sintered Cu patterns also exhibited a strong adhesion to polymer substrates, resulting in a high flexibility on bending, twisting, and adhesive taping test, which is advantageous for the fabrication of flexible LED devices (Figure 17c).

Li et al. fabricated a mixed ink of Cu nanoparticles with a diameter of 60–100 nm and a CuF-3-dimethylamino-1,2-propanediol (DMAPD) complex [137]. They treated Cu nanoparticles with an absolute ethanol solution of lactic acid (10 wt%) to remove the surface oxide layer. Due to the antioxidative property of the surface-treated Cu nanoparticles and CuF–DMAPD complex, the mixed ink could be stored under an air atmosphere for a long time. The low resistivity of 18 μΩ cm was obtained on the sintered Cu film at 200 °C for 1 h under an N_2_ atmosphere from the mixed ink containing Cu nanoparticles (0.15 g) and CuF–DMAPD complex (0.092 g). Tam et al. also developed a screen-printable mixed paste for printed electronics, where Cu precursor (Cu hydroxide and Cu formate) and Cu flakes were used as Cu sources in the paste [138]. The sintered Cu film from the mixed paste exhibited a lower resistivity (21 μΩ cm) on a glass slide at 200 °C for 3 min under an N_2_ atmosphere.

Kawaguchi et al. fabricated a mixed paste of CuF–AmIP complex and Cu flake for the production of conductive Cu films on porous cellulose paper [139]. The sintered Cu film from the mixed paste exhibited a lower resistivity (~110 μΩ cm) on porous cellulose paper than that (~970 μΩ cm) on nonporous PI film at 150 °C for 1 h under an N_2_ atmosphere. The capillary force action mechanism for the lower resistivity on cellulose paper was proposed: when the composite MOD ink was deposited on the paper, the self-reductive Cu complex partly penetrated the paper due to capillary forces, producing condensed Cu flake films on its surface (Figure 17d). In addition, a binder effect was caused by the Cu particles decomposed from the self-reductive Cu complex connected to the Cu flakes. Sakurai et al. further modified the mixed paste of the CuF–AmIP complex and Cu flake for the production of highly conductive Cu films on cellulose paper for air atmosphere-sintering [140]. The Cu flakes were immersed in formic acid to remove the surface oxide layers from the Cu flake. The mixed ink of CuF–AmIP complex, formic acid-treated Cu flake, and oxalic acid additive (0.5 wt%) produced conductive Cu films with a resistivity of 70 μΩ cm on cellulose paper at a low temperature of 100 °C for 15 s in the air.

Mixed pastes of MOD and Cu particles have also been utilized for IPL sintering methods. Chung et al. used mixed inks of Cu salts and Cu nanoparticles (10–70 nm in diameter, oxide thickness >2 nm) and sintered them via flashlight irradiation in the air to achieve a highly conductive electrode pattern [141]. Various Cu salts (e.g., Cu(II) chloride, Cu(II) nitrate trihydrate, Cu(II) sulfate pentahydrate, and Cu(II) trifluoro acetylacetonate) were examined as Cu precursors in the mixed inks. Under the optimized condition (energy of 12 J/cm^2^, a pulse duration of 10 ms, and a pulse number of 1 using 30 wt% Cu(II) nitrate trihydrate), highly conductive Cu patterns (27.3 μΩ cm) with low porosities were obtained via the IPL sintering.

Suganuma’s group developed mixed pastes of submicron Cu particles with an average size of 250 nm and a complex of CuF and 2-amino-2-methyl-1-propanol (Cuf–AMP) for IPL sintering to fabricate highly conductive Cu films on flexible substrates (PET and PEN) [142]. They employed a two-step sintering process involving low-temperature heat welding at 140 °C under an N_2_ atmosphere for 10 min and a subsequent flashlight sinter-reinforcement process in the air. The low-temperature heating decomposed the Cuf–AMP complex into active Cu nanoparticles, covering the surface of submicron Cu particles. In the subsequent flashlight sintering, these active Cu nanoparticles can act as nano-welders to form highly conductive pathways, achieving a low resistivity of the sintered Cu patterns of 7.2, 26.5 and 15.9 μΩ cm on PI, PET, and PEN films, respectively. Moreover, the superior mechanical and environmental stabilities of sintered Cu patterns were also demonstrated by bending fatigue and oxidation resistance tests. Suganuma’s group also examined the size effect of the oxidation resistance of IPL-sintered Cu films from a mixed paste of submicron Cu particles (350- or 800-nm sizes) and the Cuf–AMP complex [143]. The oxidation resistance ability of IPL-sintered Cu films from the mixed paste depended on the size of the used Cu particles. In the condition exposed to a temperature of 180 °C, the sintered Cu films prepared from 800-nm Cu particles were more stable than those prepared from 350-nm particles, which can be attributed to larger Cu particles exhibiting higher oxidation resistance. At 220 °C, the result inverted since the loose structure with numerous large voids of the Cu films from larger particles provided sufficient space for oxygen and accelerated the breakage of conductive pathways between adjacent particles by the formation of Cu oxide layers.

More recently, Deore et al. formulated a screen-printable mixed ink from a CuF–3-(diethylamino)-1,2-propanediol (DEAPD) complex, together with a fractional amount of Cu nanoparticles [144]. The screen-printable mixed ink was formulated by mixing dry Cu(II) formate (40.2%), DEAPD (43.70%), Cu nanoparticles (0.40%), water (14.14%), glycerol (1.26%), and polyester binder (0.3%) in a planetary mixer for 30 min. This mixed ink enabled conductive Cu patterns (20−38 μΩ·cm) with 70−550-μm trace widths by IPL sintering on PI or PET substrates (Figure 18a–f). The long-term stability (<10% change in resistance) of over a month in ambient conditions and 10−70% relative humidity was also demonstrated.

A mixed ink containing formic acid-treated Cu nanoparticles and Ag nitrate (F-CuAg ink) was proposed for IPL sintering to reduce the oxidization of the sintered metal film [145]. The formic acid treatment removed the Cu oxide layer and facilitated the galvanic replacement of Ag nitrate by Cu nanoparticles. When the inks contained over 20 mol% Ag nitrates, the outer shell of the Cu nanoparticles was entirely covered with Ag. The resistance of the sintered metal film was 50 Ω/sq for the CuAg film upon exposure to an air atmosphere at 220 °C, whereas it was significantly increased to 10^6^ Ω/sq for the Cu-only ink. Li et al. fabricated a mixed paste containing formic acid-treated submicron Cu particles and an Ag–amino complex for IPL sintering [146]. The photonic sintering could produce the core-shell structures consisting of a Cu-rich phase in the core and an Ag-rich phase in the shell (Cu-Ag alloy) on flexible substrates (Figure 18g). The resulting Cu-Ag alloy electrode exhibited a high conductivity (3.4 μΩ·cm) and an ultrahigh oxidation resistance even up to 180 °C under an air atmosphere. The relative resistance of an electrode prepared from pure Cu ink rapidly increased to 24 after 64 h due to the oxidation, whereas the relative resistance of the Cu-Ag alloy electrodes remained almost stable (Figure 18h). The values of the relative resistance of the electrodes were 1.2, 1.4 and 2.2 after the 1000-cycle bending with bending radii of 5, 7 and 10 mm, respectively. The high flexibility is achievable owing to the fully densified microstructure and good adhesion between the Cu-Ag alloy electrodes and the PI substrate.

### 6.3. Cu Nanowires/Cu Nano or MOD Inks

Flexible electrical devices should maintain their electrical performances under some mechanical deformations, such as bending, twisting, and stretching. However, sintered metallic NPs films tend the strain fracture by these mechanical deformations [147]. Besides, sintered Cu films often involve porosity and micro-cracks within the film. Such defects could accelerate the initiation and growth of cracks, which could result in a lower mechanical strength of the film. Cu nanowires (Cu NWs) can improve the microstructure, conductivity, and mechanical stability of the sintered Cu films owing to their high mechanical strength and electrical conductivity [148,149] (Figure 19a–c).

Joo et al. fabricated a mixed paste of Cu NWs and Cu nanoparticles to increase the reliability of sintered Cu patterns under mechanical bending fatigue [150] (Figure 19d). The mixed paste with 5 wt% Cu NWs exhibited a resistivity of 22.77 μΩ·cm via IPL-sintering (12.5 J·cm^−2^ irradiation energy, 10 ms pulse duration, and one pulse). This resistivity was lower than that of only Cu nanoparticles or only the Cu NW ink, which were 94.01 and 104.15 μΩ·cm, respectively. Under mechanical fatigue conditions over 1000 cycles with an outer bending radius of 7 mm, the resistance change (ΔR·R_0_^−1^) of the Cu NWs/Cu nanoparticle film was much smaller (4.19) compared with 92.75 of the Cu nanoparticle film without Cu NWs.

More recently, Zhang et al. fabricated a mixed ink of Cu NWs and Cu–amine complexes (Cu–AMP, Cu–Ethy, and Cu–Hexy) to control the decomposition process and create Cu films with compact structures, high conductivity, and flexibility on flexible polymer substrates [151]. The introduction of Cu NWs not only provided a network of nucleation sites for the in situ formed Cu nanoparticles from the Cu–amine complex but also led to the bridging of these Cu nanoparticles to realize a compact structure (Figure 19e–g). The Cu film from the mixed paste of the Cu–Ethy complex and Cu NWs exhibited a low resistivity of 14.9 μΩ cm at a sintering temperature of 140 °C, which was reduced by about 45%, compared with that of the Cu film obtained from the Cu complex without Cu NWs. Besides, the obtained Cu films could maintain their high conductivity even after 1000-cycle bending tests owing to the bridge function of Cu NWs and the enhanced connections between Cu particles. The flexible LED circuits and V-shaped dipole antennas prepared from the mixed paste exhibited excellent performances owing to their outstanding conductivity and flexibility.

### 6.4. Carbon Nanotube/Cu-Based Inks

To prevent crack growth and propagation, one-dimensional (1D) nanomaterials, such as carbon nanotubes (CNTs) and Cu NWs, have been used in Cu-based inks/pastes. The CNTs are promising materials for flexible electronics owing to their unique properties, such as a high intrinsic conductivity, solution processability, and flexibility. Besides, the oxidation problem, as seen in Cu NWs, is neglectable for CNTs. Hwang et al. employed multiwalled CNTs (MWCNTs) to improve the conductivity and fatigue resistance of IPL-sintered Cu nanoparticle films [152]. The Cu nanoparticle/MWCNT (CNT length = 200 μm, 0.5 wt%) composite films were fabricated via optimized flashlight irradiation (flashlight = 15 J/cm^2^, 1 pulse, 10 ms) (Figure 20a–c), and exhibited the lowest resistivity (7.86 μΩ·cm). It was also demonstrated that the composite films had a better durability (1000-cycle bending fatigue for all cases of 7-, 10- and 15-mm bending radii) (Figure 20d) and environmental stability (85 °C and RH 85%) than Cu nanoparticles only. Additionally, CNTs connected the cracked Cu nanoparticle film and acted as a channel for electrons after the bending fatigue test (Figure 20e). Rosen et al. prepared a mixed paste of submicron Cu formate particles and single-wall CNTs (SWCNT) for IPL sintering [153]. The CNTs act as a photonic additive to induce energy absorption and also provide a conductive path in the vacant space of Cu film. The addition of 0.5 wt% CNT to the paste enhanced absorption by about 50%, and the threshold energy required to obtain a conductive pattern decreased by ~25%. Seong et al. also fabricated an ultradurable and uniform-sintered Cu electrode at 140 °C under an N_2_ atmosphere from the mixed paste of MWCNT and Cu MOD [154]. The MWCNT enhances not only the uniformity of the Cu electrode by providing a heterogeneous nucleation site, but also the durability of the electrode due to the mechanical robustness of MWCNT. With an optimum MWCNT content (1.0 wt%), the Cu/MWCNT film exhibited a resistivity of 25.31 μΩ cm. The resistance increase was only 1.66 times after the 150,000-cycle bending test with a 2.6-mm radius.

## 7. Adhesion Enhancement of Sintered Cu Films on a Flexible Substrate

The adhesion between the sintered Cu patterns and flexible substrate is essential to achieve the optimal performance and reliability in printed conductors. The adhesion of the two materials depends on the morphological and chemical properties of the interface. However, the sintered Cu film has a weak adhesive strength to a flexible polymer substrate. The addition of adhesion promoters or the surface modification of the substrate can improve the adhesion.

Lee et al. reported a Cu complex ion ink containing silane coupling agents (3-aminopropyl-trimethoxy-silane: APS) as an adhesion promoter [157]. The PI substrate was treated with oxygen plasma, and the chemical reaction between the APS in ink and the oxidized PI surface enhanced the adhesion of Cu patterns on the PI film. The sintered Cu film was fabricated from the Cu ink, including 3 wt% APS at 200 °C for 2 h in H_2_. The resultant Cu patterns exhibited not only the highest peel strength (240.3 gfmm^−1^ and ASTM 4B), but also a low resistivity (approx. 20 μΩ·cm).

Jeon et al. prepared mixed ink, including Cu nanoparticles (10–70 nm in diameter), a Cu precursor (Cu(II) nitrate trihydrate), and 3 wt% silane coupling agents (3-amino-propyltrimethoxy-silane, APS) for a two-step flashlight sintering method (preheating with 7 J/cm^2^ irradiation energy, followed by 9 J/cm^2^ as primary sintering) [155]. The sintered Cu patterns from the mixed ink exhibited the highest adhesion level (5B) (Figure 21a), the lowest resistivity (7.6 μΩ·cm), and a small resistance change (18%) after the 1000-cycle bending fatigue test (with an outer bending radius of 15 mm) (Figure 21b). These characteristics are attributed to the reaction between the APS and hydroxyl group of the PI surface and the higher surface roughness resulting from the oxygen plasma treatment of PI film.

Sakurai et al. investigated the effect of various silane coupling agents in mixed pastes, including the CuF–AmIP complex and Cu flakes, on the flexibility and environmental durability of Cu electrodes on cellulose papers [158]. They utilized four different silane coupling agents: APS, aminoethyl-aminopropyltrimethoxysilane (AEAPTMS), 3-glycidoxypropyltrimethoxysilane (GPTMS), and mercaptopropyltrimethoxysilane (MPTMS). The two amine groups of AEAPTMS have a high affinity for the Cu surface, which could strengthen the interparticle binding force within the sintered Cu electrode. As a result, a negligible resistance change was observed in the sintered Cu electrode with AEAPTMS: R/R_0_ = 1.15 after 1000 repeated bending cycles with a bending radius of 5 mm and R/R_0_ = 1.4 after exposure at 60 °C/80% RH for 7 days. Besides, Cu electrodes produced from the mixed ink with GPTMS had an improved flexibility with a negligible resistance change even after 10,000 bending cycles due to the flexible alkyl ether chain of GPTMS. The Cu electrode with GPTMS could be successfully applied to paper-based flexible finger motion sensors.

One disadvantage of the addition of adhesion promoters, such as the abovementioned silane coupling agents, is the decrease in electrical conductivity due to the remaining nonconductive promoters. Besides, silane coupling agents also tend to make the Cu electrode brittle and fragile. To address this matter, Kwon et al. employed surface modifications with a self-assembled monolayer (SAM) of (3-mercaptopropyl) trimethoxysilane on the oxygen-treated PI film to enhance the adhesion of the inkjet-printed Cu electrode from a Cu complex ion ink on PI film [156] (Figure 21c). The superior adhesion strength (1192.27 N/m) and the excellent mechanical stability on 100,000 bending cycles were demonstrated on the printed Cu films on the PI substrate. Additionally, the nanostructured SAM treatment on printed Cu prevented the formation of native oxide layers (Figure 21c). The combination of the SAM modification and IPL sintering achieved a very low resistivity of the sintered Cu film of 2.3 μΩ·cm on the PI film. Printed and skin-conformal Cu electrodes indicated a very low skin-electrode impedance (<50 kΩ) without a conductive gel and successfully measured three types of biopotentials, electrocardiograms, electromyograms, and electrooculograms (Figure 21d).

Cho et al. demonstrated an enhanced adhesion between a Cu electrode formed by IPL sintering and a chemical treatment-PI substrate without reducing the electrical conductivity [159]. The surface of the PI substrate was converted to polyamic acid via potassium polyamate formation with potassium hydroxide and hydrochloric acid. During the IPL sintering process on Cu nanoparticle inks, the polyamic acid layer could migrate into the cavities between the sintered Cu nanoparticles. Thus, an interlocking structure was formed. The increased contact area between the Cu nanoparticles and the substrate surface could dramatically improve the adhesion strength of the Cu electrode from 0B to 5B. The low resistivity of 4.17 μΩ·cm was obtained on the sintered Cu electrode on the PI.

## 8. Conclusions and Outlook

Studies on Cu-based inks/pastes have focused on oxidation resistance and low-temperature sintering and sintering methods for producing highly conductive and stable Cu patterns on various flexible substrates, and the mechanical/environmental reliability of flexible Cu films/electrodes for flexible applications in electronics. However, despite the significant progress, several issues on the widespread use of cost-effective Cu-based inks/pastes in next-generation electronic devices still exist.

Obtaining highly conductive (less than 5 μΩ·cm) Cu patterns with a high mechanical/environmental reliability on heat-sensitive flexible substrates is greatly challenging, and the IPL sintering methods may be suitable for this. Various factors (e.g., pulse duration, irradiation energy, number of pulses, and multiple heating/drying process) on the IPL sintering need to be established by optimizing individual Cu-based inks/pastes. Besides, high-resolution Cu patterns will be required to achieve the miniaturization of printed electronics by the development of ink formulations and printing methods. In numerous cases, we have to take note of the oxidation problems that can be encountered with various situations regarding Cu nanomaterials: the synthesis of Cu nanoparticles, ink/paste formulation, storage of ink/paste, sintering process, and long-term reliability of sintered Cu films. The surface and interface engineering in Cu-based inks/pastes, such as the search for protective organic/inorganic layers and surface-activatable treatments, is likely to be a pivotal approach for solving these problems. A scalable, eco-friendly, and cost-effective fabrication of Cu-based inks/pastes is necessary for practical/industrial applications. Thus, the use of a nontoxic reducing agent and large-scale one-pot synthesis of Cu particles are desirable, and in that sense, MOD inks may be suitable for these viewpoints, except for the low Cu content.

Finally, we mention about future outlooks for the development tendency in Cu-based inks/pastes and potential applications of flexible Cu-based electrodes. Cu-based inks/pastes are recognized as having a low reliability due to oxidation. As a result, there are few reports on the application of Cu-based inks/pastes on flexible electronics applications, compared with those of Ag-based inks/pastes and carbon nanotube-(CNT) or graphene-based inks/pastes [3,4,5,6,7,8,9,10,11,12,13,14,15,16,17,18,19,20,21]. However, the recent development of Cu-based inks/pastes described in this review would allow a real application of flexible devices such as smart robotics, electronic skin, optoelectronics, human motion, health monitoring systems, and human–machine interactive systems in the future. Recently, human-friendly stretchable electronics, combined with analytical biosensing, are extending the concept of healthcare and enabling much more accurate and comfortable monitoring in real-time [160,161,162]. However, “metal”-based electronics are limited to the stretchability compared to “organic”-based electronics. Metal/elastomer composite-based inks/pastes would be attractive for the fabrication of stretchable electronics. Another method to design the stretchable electronics is the use of gallium-based liquid metals (LMs). The combination of the fluidity and metallic nature with a high heat/electrical conductivity of LMs is attractive for stretchable electronics applications [163,164]. The mixture of LMs and Cu particles (Cu–EGaIn) offers high electrical conductivity, excellent stretchability, and printability [165,166,167]. These advanced characters would be promising pastes for next-generation stretchable electronics in the future.

High-power semiconductor devices, such as SiC and GaN, with wideband gaps, are expected to function for an extended period at temperatures above 200 °C, even 350 °C, which is too high for the traditional die-attach materials that are currently used widely. Cu-based inks/pastes have high potentials for low-temperature Cu–Cu bonding for high-power semiconductor devices as the next-generation die-attach nanomaterials [45,168,169,170,171,172,173,174,175]. Three-dimensional (3D) printed electronics is now an emerging field that enables the fabrication of nonflat or 3D molding [176,177]. Metal 3D technologies can produce metal prototypes with the freedom of design linked to additive manufacturing. At present, direct metal laser sintering (DMLS) is employed for the primary sintering process for 3D metal printing using metal powders. The development of Cu-based/pastes for metal 3D printing is a great challenge to produce highly conductive 3D architectures using a cost-effective heating process at low-temperatures [178,179,180].

We expect the further study of metal-based inks/pastes to promote the development and application of next-generation electronics, which are likely to have a significant impact on our society.

## Figures and Tables

**Figure 1 nanomaterials-10-01689-f001:**
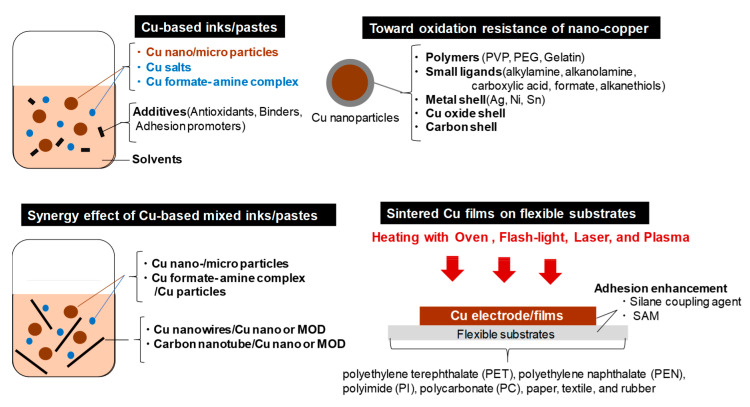
Strategies for Cu oxidation resistance and low-temperature sintering by the surface and interface designs in Cu-based conductive inks/pastes.

**Figure 2 nanomaterials-10-01689-f002:**
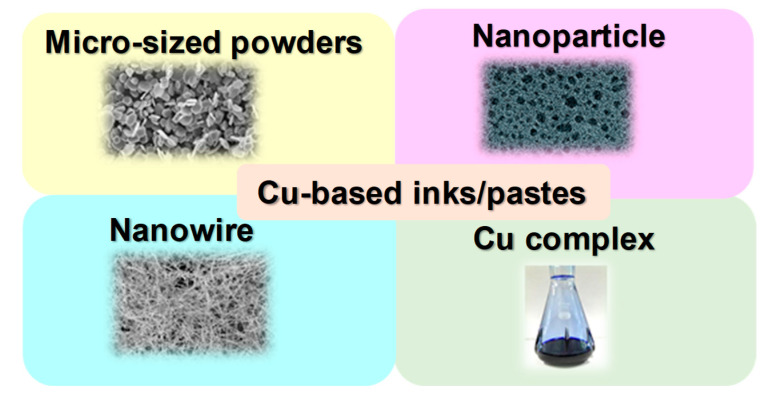
Categories of Cu-based inks/pastes.

**Figure 3 nanomaterials-10-01689-f003:**
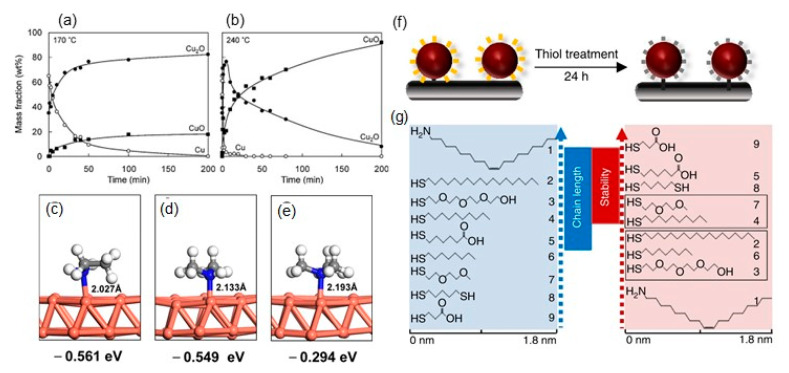
Mass fractions of Cu, Cu_2_O, and CuO in nanoparticles oxidized at (**a**) 170 °C and (**b**) 240 °C. Reprinted with permission from reference [31], Copyright Elsevier, 2011. Adsorption property calculation of (**c**) primary amine, (**d**) secondary amine, and (**e**) tertiary amine groups of polyethylenimine (PEI) with a Cu (111) surface. Reprinted with permission from reference [38], Copyright American Chemical Society, 2018. (**f**) Schematic representation of the ligand exchange process where oleylamine (OAM) attached to the CuNPs is replaced by incoming thiol ligand. (**g**) Molecular structures of OAM and the thiols used for ligand exchange and capping of Cu nanoparticles sorted according to the chain length and stability in the air. Reprinted with permission from reference [39], Copyright Springer Nature, 2017.

**Figure 4 nanomaterials-10-01689-f004:**
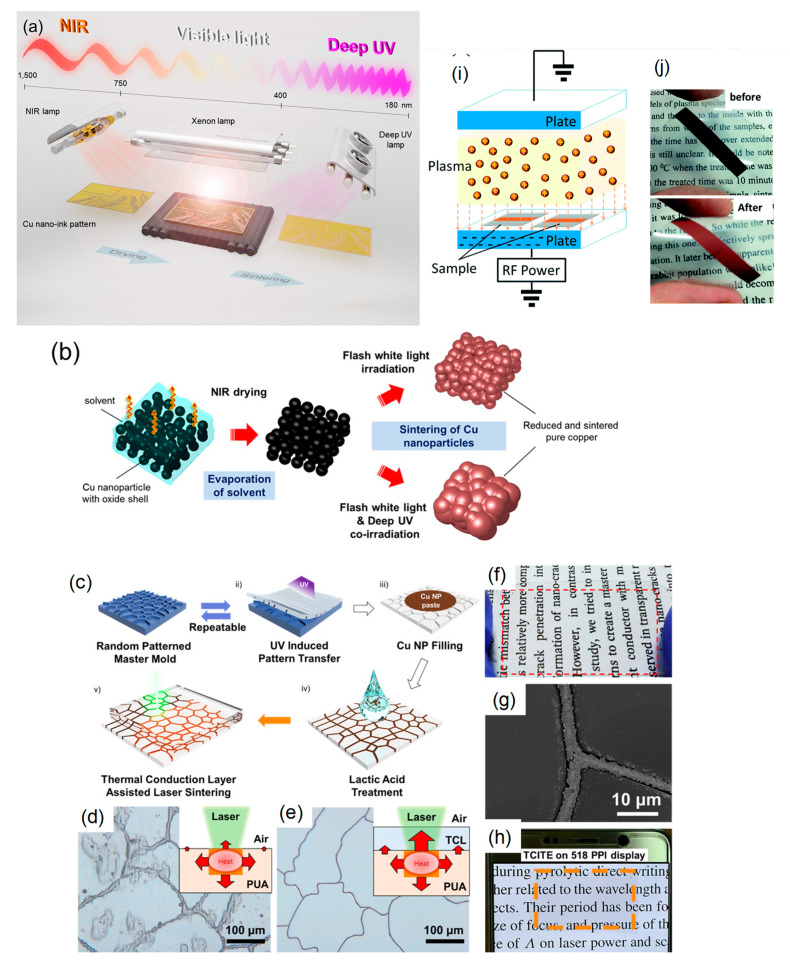
(**a**) Schematics of the photonic drying and sintering process of Cu nano ink, using near-infrared (NIR), flash white light, and deep UV. (**b**) The morphological change of Cu nanoparticles in nano ink during the photonic drying and sintering process. Reprinted with permission from reference [51], Copyright Springer Nature, 2016. (**c**) Schematic fabrication process: a randomly patterned master mold is used for pattern transfer onto the polyethylene terephthalate (PET) film using UV-curable epoxy. Cu nanoparticles (Cu NPs) are filled in the trench. The nanoparticles are treated with lactic acid and sintered by the thermal conduction layer (TCL)-assisted laser sintering process. The microscopic image of the (**d**) conventional laser-sintered electrode and (**e**) TCL-assisted laser-sintered electrode (TCITE). The inset images depict relative heat transfer rates between materials. (**f**) Highly transparent fabricated electrode. (**g**) Narrow line width (~2.4 μm) of the fabricated electrode. (**h**) TCITE overlaid on high-resolution (518 ppi) display. Reprinted with permission from reference [55], Copyright American Chemical Society, 2019. Schematic diagram of the plasma process (**i**) and Cu pattern photos before and after plasma treatment (**j**). Reprinted with permission from reference [56], Copyright Royal Society of Chemistry, 2015.

**Figure 5 nanomaterials-10-01689-f005:**
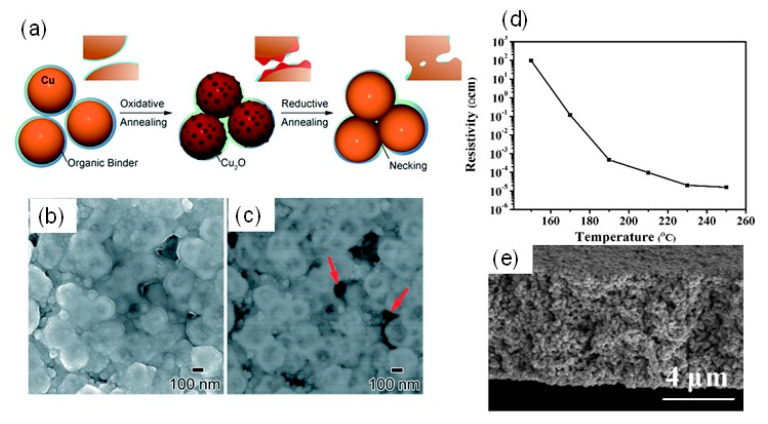
(**a**) Schematic of the necking and connection formation between the Cu fine particles using our two-step annealing process. SEM images of the Cu particle layer surface after oxidative annealing. (**b**) An SE (secondary electron) image and (**c**) a BEI-COMPO image with the red arrows marking the areas of organic material. Reprinted with permission from reference [58], Copyright Royal Society of Chemistry, 2015. (**d**) Electrical resistivity of Cu conductive patterns annealed at different temperatures. (**e**) Cross-section SEM image of Cu conductive patterns sintered at 250 °C. Reprinted with permission from reference [62], Copyright American Chemical Society, 2014.

**Figure 6 nanomaterials-10-01689-f006:**
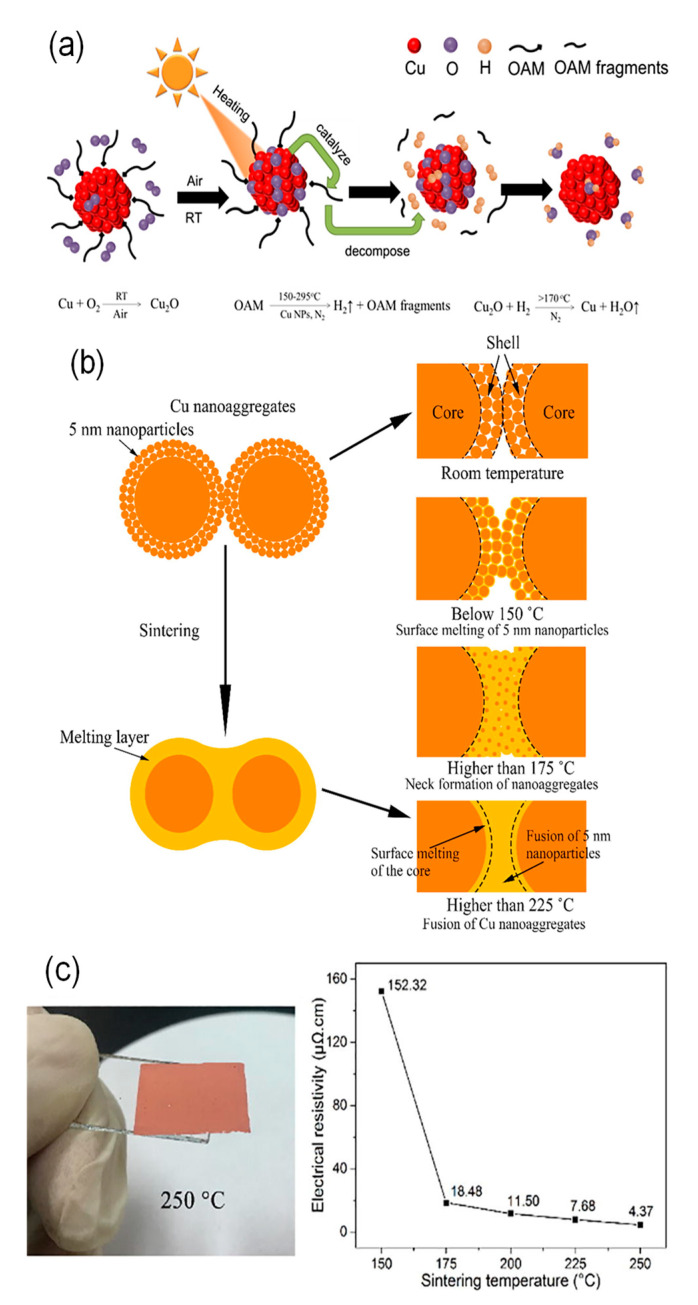
(**a**) Schematic of self-reducible oleylamine (OAM)-protected Cu nanoparticles for conductive ink. Reprinted with permission from reference [65], Copyright American Chemical Society, 2018. (**b**) The schematic diagram of the sintering mechanism of Cu nanoaggregates. (**c**) Photograph of Cu film after sintering at 250 °C. (**d**) Electrical resistivity changes of the sintered Cu films. Reprinted with permission from reference [69], Copyright Springer Nature, 2018.

**Figure 7 nanomaterials-10-01689-f007:**
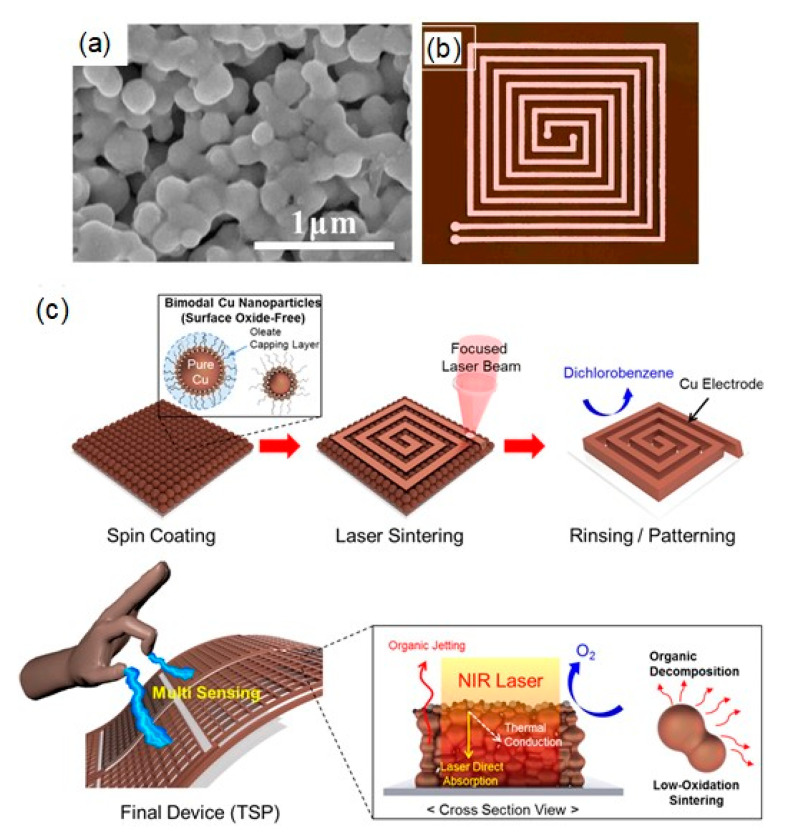
(**a**) SEM image of prepared ink annealed at 240 °C. (**b**) Conductive Cu patterns on polyimide (PI) substrate. Reprinted with permission from reference [73], Copyright Springer Nature, 2018. (**c**) Schematic illustrations showing the patterning process using a transversally extended laser plasmonic (TLP) welding for Cu nanoparticle (NP) assemblies and its final device. Reprinted with permission from reference [80], Copyright American Chemical Society, 2016.

**Figure 8 nanomaterials-10-01689-f008:**
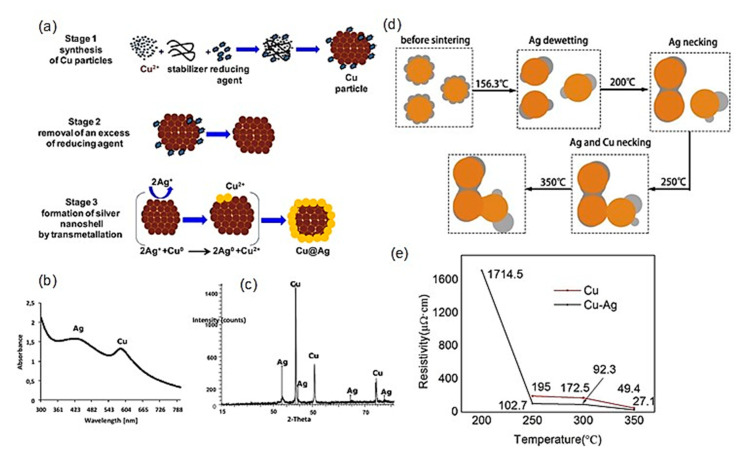
(**a**) Schematic of the three-stage synthesis of Cu@Ag particles using the transmetalation method. (**b**) The absorption spectrum of an aqueous dispersion of Cu@Ag particles and (**c**) XRD pattern of Cu@Ag particles. Reprinted with permission from reference [85], Copyright Elsevier. 2017. (**d**) Schematic of the sintering mechanism for Cu-Ag core-shell nanoparticles. (**e**) Electrical resistivity of the conductive films containing Cu nanoparticles and Cu-Ag core-shell nanoparticles. Reprinted with permission from reference [86], Copyright Elsevier, © 2017.

**Figure 9 nanomaterials-10-01689-f009:**
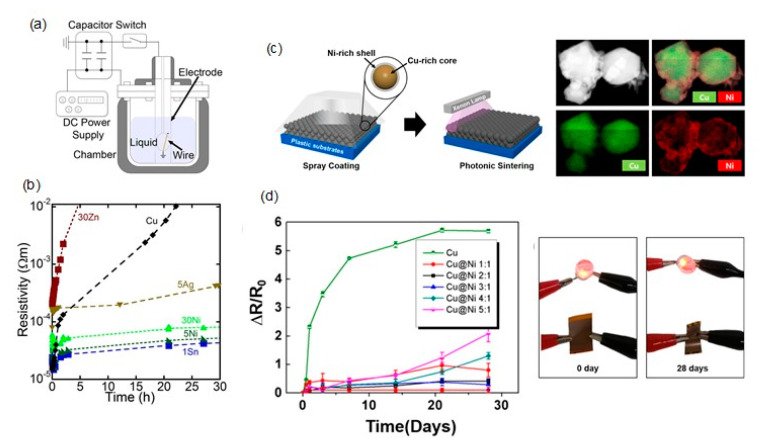
(**a**) Illustration of the wire exploder to produce alloy nanoparticles. (**b**) Time-dependent conductivity attenuation during exposure to 85 °C and 85% relative humidity (RH). Reprinted with permission from reference [91], Copyright Springer Nature, 2015. (**c**) Schematic illustration of the fabrication of large-area Cu@Ni core-shell NP-based electrodes on a polymeric substrate and high-resolution transmission electron microscopy (HRTEM) image and elemental distribution maps of Cu@Ni core-shell NPs. (**d**) A plot of relative resistivities versus time for various NP-based electrodes under 85 °C/85% relative humidity aging and photographs showing the stable resistance of a Cu@Ni NP-based electrode before and after humidity aging test. Reprinted with permission from reference [94], Copyright American Chemical Society, 2018.

**Figure 10 nanomaterials-10-01689-f010:**
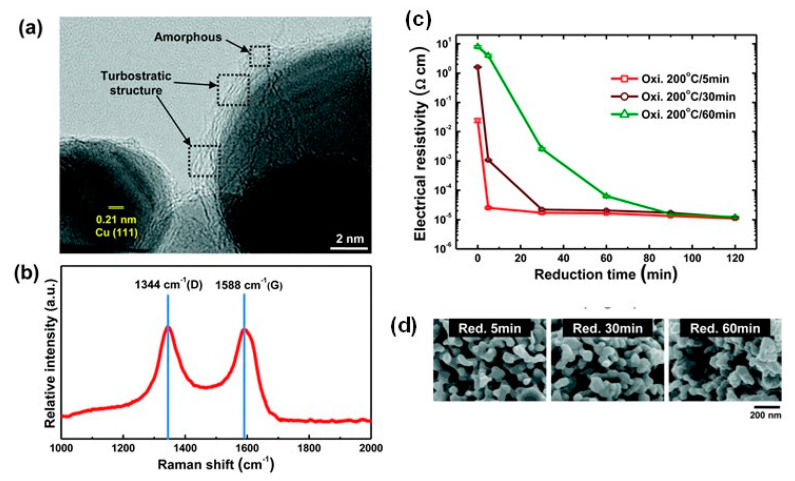
(**a**) Representative high-magnification TEM image showing the core-shell nanostructure of Cu@C nanoparticles and (**b**) Raman spectrum of Cu@C nanoparticles. (**c**) Variation of electrical resistivity of the films oxidized at 200 °C for 5, 30 and 60 min as a function of isothermal reduction time under an Ar/10% H_2_ atmosphere. (**d**) SEM images showing the morphologies with an increase in isothermal reduction time for the film oxidized for 5 min at 200 °C. Reprinted with permission from reference [97], Copyright Royal Society of Chemistry, 2015.

**Figure 11 nanomaterials-10-01689-f011:**
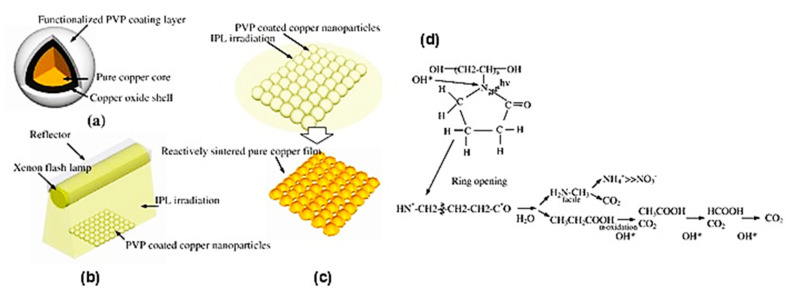
Schematic of reactive sintering using intense pulsed light (IPL): (**a**) polyvinyl pyrrolidone (PVP)-coated Cu nanoparticles; (**b**) IPL irradiation using the xenon flash lamp; (**c**) reactive sintering of Cu nanoparticles by IPL irradiation. (**d**) The photodegradation phenomena of PVP resulting in the generation of alcohol or acid. Reprinted with permission from reference [100], Copyright Springer Nature, 2011.

**Figure 12 nanomaterials-10-01689-f012:**
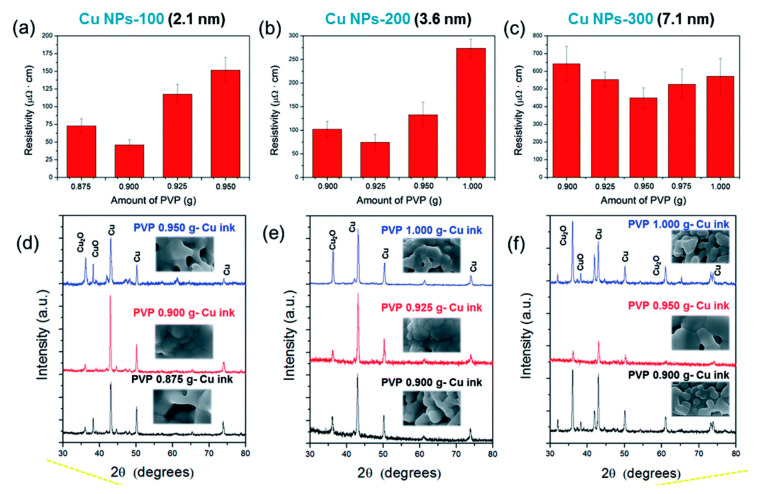
Optimization of PVP amount for reducing and sintering of Cu NP ink with different oxide shell thickness. (**a**–**c**) The resistivity of Cu nano ink films with varying amounts of PVP to the thickness of Cu oxide shell. (**d**–**f**) XRD patterns of the flashlight-sintered Cu nano ink films before (black line) and after (red line) optimization of the PVP amount (irradiation energy: 12.5 J cm^−2^, pulse duration: 10 ms, pulse number). Reprinted with permission from reference [104], Copyright Royal Society of Chemistry, 2017.

**Figure 13 nanomaterials-10-01689-f013:**
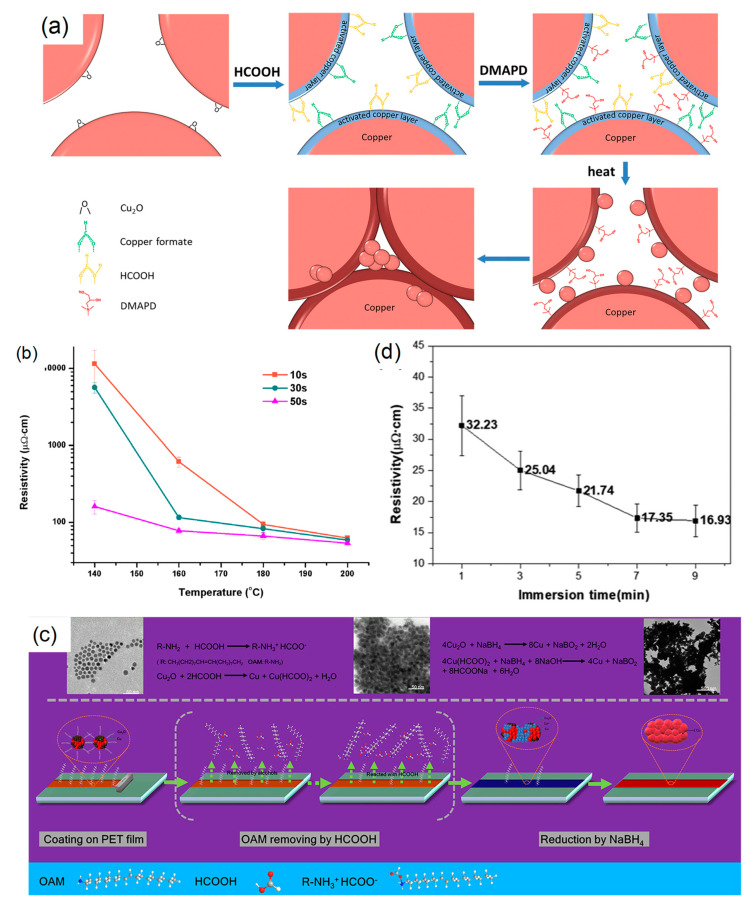
(**a**) Schematic of the low-temperature sintering process of Cu–HCOOH–3-dimethylamino-1,2-propanediol (DMAPD) composite. (**b**) Electrical resistivity of the Cu films of Cu–HCOOH–DMAPD (10:2:2) sintered at various temperatures in ambient air. Reprinted with permission from reference [112], Copyright Springer Nature, 2019. (**c**) Schematic illustration of the chemical sintering mechanism using folic acid and NaBH_4_. (**d**) Final resistivity of the metal layer treated with 10 vol% HCOOH in methanol, followed by 0.75 wt% NaBH_4_ immersion for different periods. Reprinted with permission from reference [121], Copyright American Chemical Society, 2020.

**Figure 14 nanomaterials-10-01689-f014:**
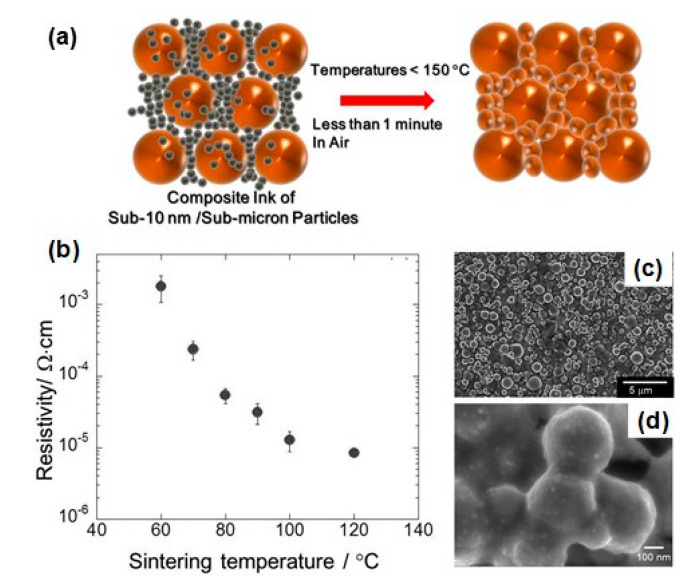
(**a**) Schematic diagram showing the sintering mechanism of the composite inks. (**b**) The electrical resistivity of Cu films from the composite ink with the 3:1 mixture (wt%) of Cu particles and AmIP−Cu NPs sintered at various temperatures for 60 min under N_2_ flow. (**c**,**d**) SEM images of the Cu film prepared at a sintering temperature of 80 °C. Reprinted with permission from reference [125], Copyright American Chemical Society, 2017.

**Figure 15 nanomaterials-10-01689-f015:**
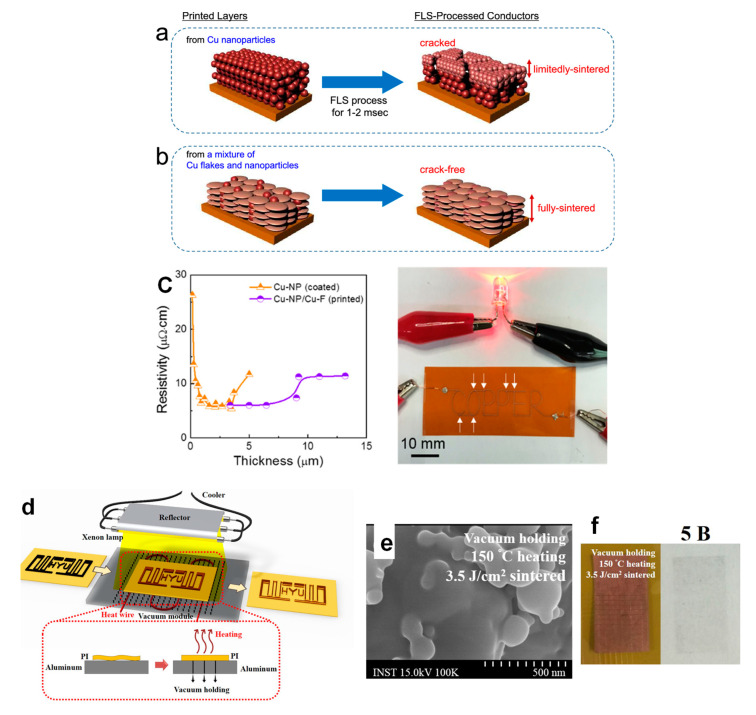
(**a**) Microstructural evolution of sintered Cu conductors prepared from (**a**) nanoparticles and (**b**) a mixture of flakes and nanoparticles. (**c**) Variation in resistivity as a function of the thickness of Cu conductors. The Cu–NP-based conductors were spray-coated without forming patterned structures, and the Cu-F/Cu-NP mixture-based conductors were printed directly with the generation of patterned structures. Photograph of the printed Cu circuit connected with battery and light-emitting diode (LED). Reprinted with permission from reference [126], Copyright American Chemical Society, 2019. (**d**) Schematic of IPL sintering system with a vacuum heating module. Inset figure expresses substrate heating and vacuum holding. (**e**) SEM images of Cu NP/MP-ink films:3.5 J/cm^2^ IPL-sintered with vacuum holding and 150 °C heating. (**f**) Adhesion strength test results of Cu NP/MP-ink films:3.5 J/cm^2^ IPL-sintered with vacuum holding and 150 °C heating. Reprinted with permission from reference [127], Copyright Elsevier, 2019.

**Figure 16 nanomaterials-10-01689-f016:**
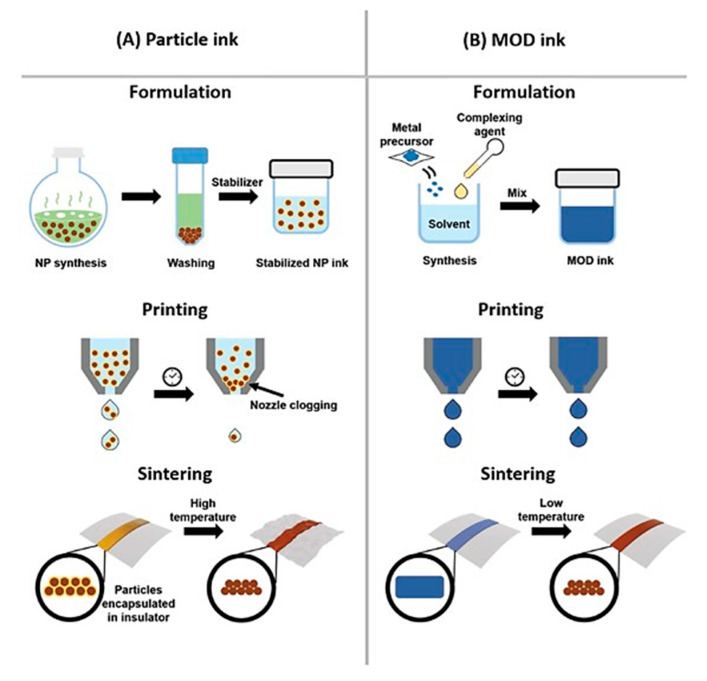
Preparation processes and working mechanisms of (**A**) particle ink and (**B**) metal-organic decomposition (MOD) ink. Formulation, printing, and sintering processes for both types of ink and possible problems that may arise during printing (nozzle clogging of particle ink) and sintering (substrate degeneration upon sintering at a high temperature for particle ink) are depicted in the simplified illustration. Reprinted with permission from reference [131], Copyright WILEY-VCH Verlag GmbH & Co, 2019.

**Figure 17 nanomaterials-10-01689-f017:**
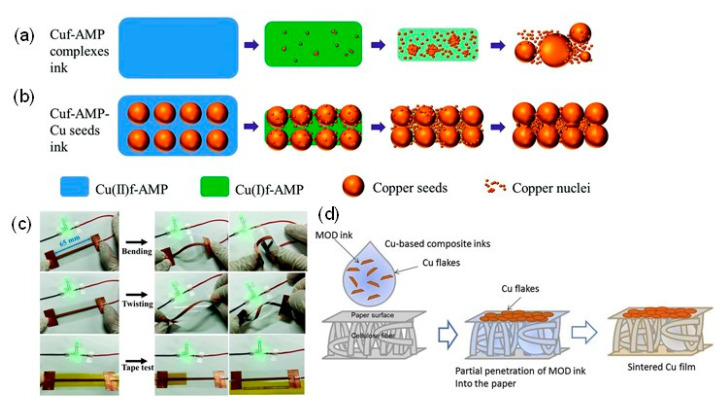
Schematic illustration of the nucleation and sintering behaviors of (**a**) Cu formate (CuF) and 2-amino-2-methyl-1-propanol (AMP) complex ink and (**b**) CuF–AMP–Cu particle mixed ink. (**c**) Photographs of the LED circuit with a 65-mm-long printed conductive pattern on PET during the bending, twisting, and adhesive tape tests. Reprinted with permission from reference [136], Copyright Royal Society of Chemistry, 2016. (**d**) Schematic of the capillary force-assisted formation of a condensed conductive Cu film on paper from the composite MOD ink. Reprinted with permission from reference [139], Copyright Elsevier, 2017.

**Figure 18 nanomaterials-10-01689-f018:**
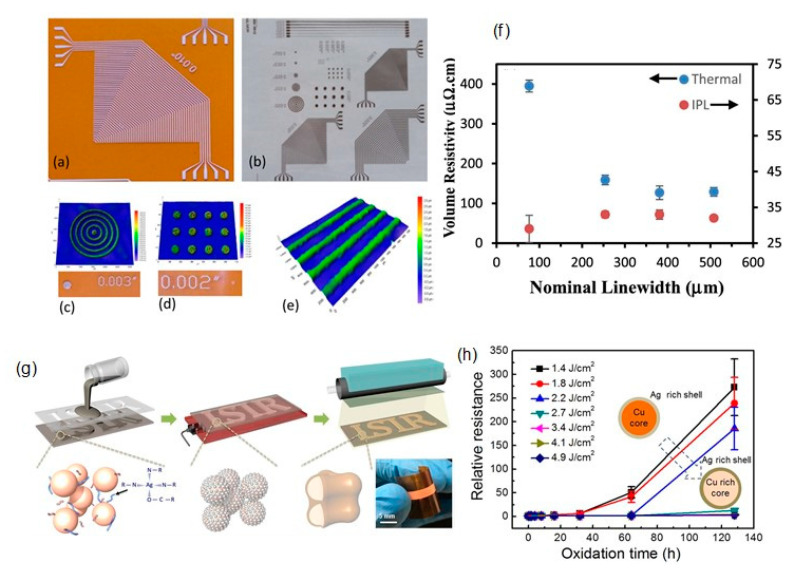
Photographs of thermally sintered Cu traces on (**a**) Kapton and (**b**) PET produced from screen printing of MOD ink. Test pattern comprising a 10-cm-long straight line ranging from ~70- to 550-μm width, bent traces, concentric circle, and dots is shown on PET (**b**). The 3D optical profilometry images of concentric circle, dots, and lines are presented in (**c**–**e**), respectively. (**f**) Volume resistivity of screen-printed Cu traces as a function of nominal line width from Cu molecular ink and IPL sintered on PET. Reprinted with permission from reference [144], Copyright American Chemical Society, 2019. (**g**) Schematic diagram: mask printing and fabrication of Cu-Ag alloy electrodes or circuits by low-temperature procuring and rapid sintering under an air atmosphere. (Inset) Photograph of flexible Cu-Ag alloy electrode on a PI substrate. (**h**) The oxidation resistance of Cu-Ag solution alloy electrodes after different energies of photonic sintering. Relative resistance is plotted as a function of oxidation time at 180 °C in the air. Reprinted with permission from reference [146], Copyright American Chemical Society, 2017.

**Figure 19 nanomaterials-10-01689-f019:**
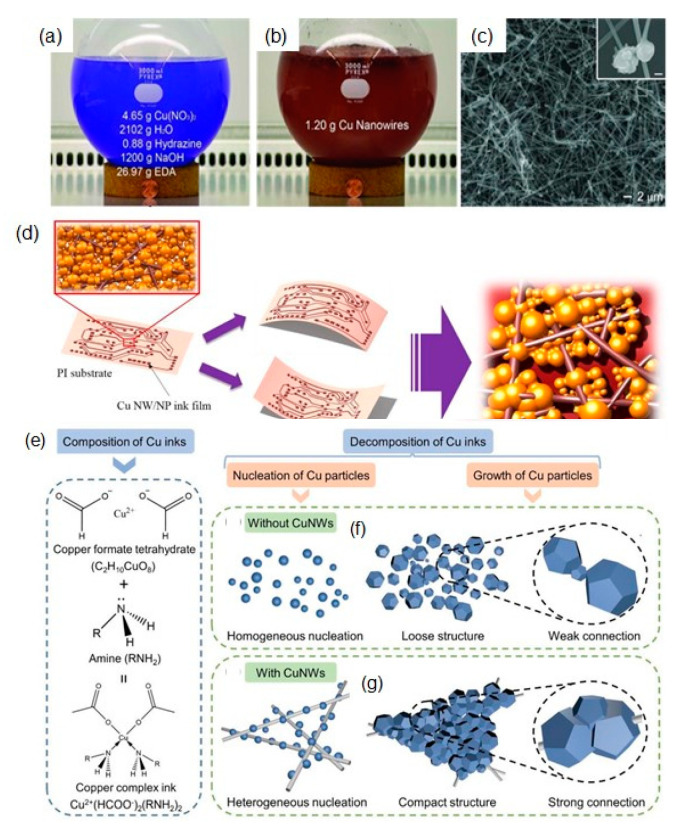
Schematic diagram of the reaction flask (**a**) before the synthesis and (**b**) after the growth of Cu nanowires (Cu NWs) at 80 °C for 1 h. (**c**) SEM image of the Cu NW product. Reprinted with permission from reference [148], Copyright WILEY-VCH, 2010. (**d**) A schematic illustration of the improved reliability of flashlight-sintered Cu NW/nanoparticle ink film under outer bending (tension) and inner bending (compression) conditions. Reprinted with permission from reference [150], Copyright American Chemical Society, 2015. (**e**–**g**) Schematic illustration showing the composition and molecular structure of Cu complex inks and the decomposition process of pure Cu complex inks and Cu complex inks with the addition of Cu NW networks. Reprinted with permission from reference [151], Copyright WILEY-VCH, 2020.

**Figure 20 nanomaterials-10-01689-f020:**
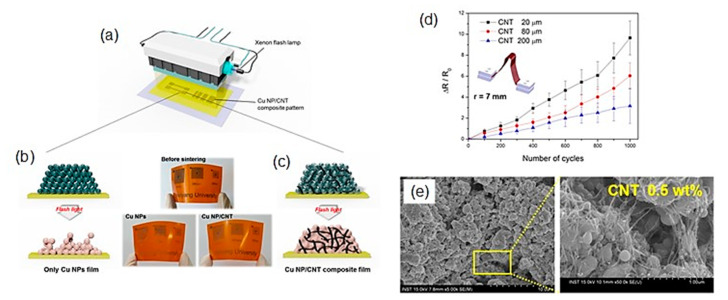
Schematics of the flashlight sintering system (**a**) and the shape change comparison of Cu nanoparticle film only (**b**) and Cu nanoparticle/carbon nanotube (CNT) composite film (**c**) after flashlight irradiation. (**d**) Results of outer bending fatigue test of Cu nanoparticle/CNT composite films with a bending radius of r = 7 mm. (**e**) The SEM images of the fatigue-tested Cu nanoparticle/CNT composite films after 1000-cycle bending fatigue test of 0.5 wt% CNTs. Reprinted with permission from reference [152], Copyright American Chemical Society, 2015.

**Figure 21 nanomaterials-10-01689-f021:**
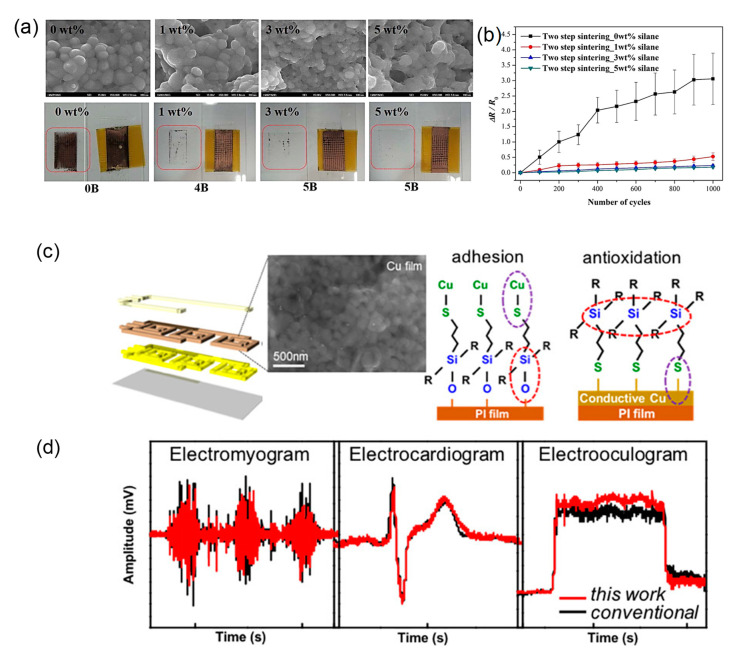
(**a**) SEM images of the sintered Cu pattern to the weight fraction of the silane (0, 1, 3 and 5 wt%) and the results of the adhesion test. (**b**) Fatigue characteristics of the Cu pattern to the weight fraction of the silane. Reprinted with permission from reference [155], Copyright Elsevier, 2016. (**c**) Nanostructured SAM surface treatment for the improvement of adhesion strength between the printed Cu and PI film using robust functional bonding of silanol and Cu−S (adhesion). Nanostructured SAM treatment on printed Cu to block Cu oxidation (antioxidation). (**d**) Recorded biopotentials by the skin-mounted electrodes, including electrophysiological monitoring of electromyograms(EMG), with three gestures. Reprinted with permission from reference [156], Copyright American Chemical Society, 2018.

**Table 1 nanomaterials-10-01689-t001:** Summary of surface designs by polymers.

Metal Source	Protective Agents	Sintering Process	Resistivity (μΩ·cm)	Ref.
Cu nanoparticle (NPs) /particle size of 30–200 nm	PVP	300 °C/air (95%)-formic acid (5%)/15 min	36 on glass	[45]
Cu NPs/140 nm	PVP	250 °C/Ar (96%)–H_2_ (4%)/120 min	17.0 on PI	[46]
Formic acid-treated Cu NPs/30 nm	PVP	260 °C/N_2_ (95%)–H_2_ (5%)/60 min	6.1 on glass	[47]
Cu NPs with 1.6 nm CuO layer/65 ± 3 nm	PVP	250 °C/formic acid vapor/60 min 250 °C/oxalic acid vapor/60 min	2.30 on glass 3.82 on glass	[49]
Cu NPs with oxide layer of more than 2 nm/20–30 nm	PVP	IPL combined with NIR and deep UV in air	7.62 on PI	[51]
Cu NPs, oxygen <3%/<100 nm	PVP	IPL with a bandpass filter (from 500 to 600 nm)	6.97 on PI	[52]
Cu NPs/70 nm	PVP	sequential low-pressure drying, near-infrared sintering, and IPL reduction	7.9 on PI	[53]
Cu NPs/< 50 nm	PVP	Laser 532 nm in the air; Selective laser sintering	Below 1 Ω/sq	[54]
Cu NPs/40 nm	PVP	Thermal conduction layer-assisted IPL; metal grid transparent conductors	4.7 Ω/sq	[55]
Cu particle/2.5–6.5 mm Cu particles/250–700 nm	PVP	200 W Plasma (argon and H_2_ (<10 wt%))/10 min	15.9 on PET	[56]
Cu particle/130 nm	Gelatin	Annealing at 200 °C in air for more than 3 h and subsequently reduced at 200 °C in 3% H_2_ gas for 3 h	8.2 on Al_2_O_3_ substrate	[58]
Cu particles/135 ± 30 nm	PEG2000	250 °C/N_2_/30 min	15.8 on PI	[62]

**Table 2 nanomaterials-10-01689-t002:** Summary of surface designs by small organic ligands.

Metal Source	Protective Agents	Sintering Process	Resistivity (μΩ·cm)	Ref.
Cu NPs/10.6 nm	Oleylamine (OAM)	300 °C/N_2_/60 min; self-reduction with OAM; IPL; self-reduction ability	84.2 on PI 25.3 on PI	[65]
Cu NPs	Oleylamine (OAM)	400 °C/N_2_/60 min; self-reduction with OAM	23 on glass	[66]
Cu NPs/3.5 ± 1 nm	1-amino-2-propanol (AmIP)	150 °C/N_2_/15 min; self-reduction with AmIP	30 on PET	[67]
Cu NPs/4.4 nm	2-amino-1-butanol (AB)	150 °C/N_2_/60 min; self-reduction with AB	52 on PI	[68]
Cu nano-aggregates covered with 5 nm Cu NPs	1-amino-2-propanol (AmIP)	250 °C/Ar (95%)–H_2_ (5%)/60 min	4.37 on glass	[69]
Cu NPs/<10 nm	Lactic acid	200 °C/N_2_/60 min; self-reduction with lactic acid	9.1 on glass	[70]
Cu NPs/50 nm	Ascorbic acid	240 °C/N_2_/40 min	16.2 on PI	[73]
Mixture of Cu NPs and Cu particles	Citric acid	300 °C/Ar/60 min	8.2 on glass	[75]
Cu NPs/61.7 nm	Citric acid	200 °C/Ar (98%)–H_2_ (2%)/60 min, closely packed Cu nanoparticle films by capillary immersion force	7.6 on glass	[76]
Cu NPs with bimodal distribution of 42.3 and 108.3 nm	Oleic acid	250 °C/Ar (90%)–H_2_ (10%)/120 min	3.9 on glass	[77]
Cu NPs with bimodal distribution of 42.3 and 108.3 nm	Oleic acid	IPL	6.7, 19.1 and 51.2 on PI, PES, and PET	[78]
Cu NPs/90 nm	Oleic acid	IPL combined with ultrathin plasmonic optical/thermal barrier	0.24 Ω/sq, 22.6 on PET	[79]
Cu NPs/<110 nm	Oleic acid	1046 nm NIR laser 532 nm green laser	4.6 on glass38.9 on glass	[80]
Cu NPs/100 nm	1-octanethiol	230 °C/H_2_/8 h	5.4 on glass	[82]
Cu NPs/106 ± 16 nm	1-octanethiol	IPL	24 on PI	[83]

**Table 3 nanomaterials-10-01689-t003:** Summary of surface designs by core-shell structure.

Metal Source	Protective Agents	Sintering Process	Resistivity (μΩ·cm)	Ref.
Cu@Ag NPs/about 1 mm Cu core with 20 nm Ag shell	PAA	250 °C/N_2_/15 min	11.34 on glass	[85]
Cu@Ag NPs with Ag:Cu atomic ratio of 1:10/50 nm	PVP	250 °C and 350 °C/H_2_ (2%)/60 min	102.7 and 27.1 on glass	[86]
Cu@Ag NPs with Ag:Cu atomic ratio of 1:4/3–50 nm	CTAB	150 °C/N_2_/60 min	13.8 on photo-paper	[87]
Mixture of Cu@Ag particles with a diameter of 1.7 mm and Ag NPs with 60 nm	PVP and CTAB	160 °C/Ar/120 min	5.0 on photo-paper	[88]
Cu-Ag alloy nanoflakes/700 nm long and 500 nm wide	PVP	150 °C/N_2_/120 min	37.5 on PI and PET	[89]
Cu5Ag5 alloy NPs /10.95 nm	Oleic acid and oleylamine	250 °C/N_2_ (95%)–H_2_ (5%)/180 min	58 on photo-paper	[90]
Cu-Ni alloy NPs by Wire Explosion	−	200 °C/H_2_/30 min	599 on glass (95Cu-5Ni) 857 on glass (70Cu-30Ni)	[91]
CuNi (55Cu-45Ni) alloy NPs/100 nm	PVP	350 °C in vacuum furnace/120 min/chemical reduction process	400 on PI	[92]
Cu@Ni NPs (Cu:Ni = 8:1)	PVP	350 °C	27	[93]
Cu@Ni NPs (Cu:Ni = 5:1)/bimodal size distribution of 56.3 and 146.8 nm	Oleicacid and oleylamine	IPL	52, 84 and 900 for PI, PES and PEN.	[94]
Cu and Cu10Sn3 core-shell NPs (5 wt%)/20–60 nm	Oleic acid	IPL	12.2 on PI	[95]
Cu, Ni and Cu10Sn3 core-shell NPs (30 wt% Ni NPs)	Oleicacid and oleylamine	IPL	49 on PI	[96]
Cu@carbon NPs/20.6 nm	Defective carbon of 1.0 nm	200 °C for 10 min in air and subsequently reduction process under H_2_ (10%)	25.1 on PI	[97]
Cu@graphene NPs/52 ± 9 nm	Multilayer graphene of 3.0 nm	No thermal annealing	170 on PI	[99]

**Table 4 nanomaterials-10-01689-t004:** Summary of surface designs by Cu oxide.

Metal Source	Protective Agents	Sintering Process	Resistivity (μΩ·cm)	Ref.
CuO NPs/10−300 nm (Novacentrix Metalon ICI-002HV)	N.A	IPL	3.1 on CaCO_3_ precoating paper	[101]
CuO NPs/10−300 nm (Novacentrix Metalon ICI-002HV)	N.A	IPL	335 mΩ/sq on paper	[102]
CuO NPs/10−300 nm (Novacentrix Metalon ICI-002HV)	N.A	IPL	10 on PET	[103]
Cu NPs with an oxide shell of 3.6 nm/20–50 nm	PVP	IPL	43.83 on PI	[104]
Cu NPs with an oxide shell of more than 2 nm/40 nm	PVP	IPL (preheating and main sintering)	3.81 on PI	[105]
Cu NPs (100 nm) with an oxide shell of 20 nm and CuO NPs (50 nm); Cu/CuO weight ratio of 1/80	PVP	IPL	6.5 on PET	[106]
Cu_2_O NPs/5.1 ± 0.4 nm	L-ascorbic acid	IPL	4.2 on PI	[107]
Cu NPs from CuO particles by chemical reduction/280 nm	Octylamine	Annealing at 250 °C in air for 4 h, and subsequently reduced at 250 °C in 3% H_2_ in N_2_ gas for 3 h	7.8 on Al_2_O_3_ substrate	[108]

**Table 5 nanomaterials-10-01689-t005:** Summary of surface designs by surface activation.

Metal Source	Protective Agents	Sintering Process with Surface Activation Process	Resistivity (μΩ·cm)	Ref.
Cu flakes/50 mm width, 1 mm thickness	N.A.	120 and 140 °C/reduction-assisted sintering with ethanol vapor	about 1000 and 100 on PEN	[111]
Cu particles/820 nm	N.A.	200 °C for 50 s in air/surface modification with Cu–HCOOH–DMAPD complex (DMAPD: 3-dimethylamino-1,2-propanediol)	54 on PI	[112]
Cu flakes/50 mm width, 1 mm thickness	N.A.	Room temperature under vacuum-drying (No thermal annealing)/reduction-assisted sintering with immersion in ascorbic acid solution	6900, 7400, 7700, 5700, and 3270 on PI, PEN, PC, PMMA, and PET	[118]
Cu NPs/<10 nm	1-amino-2-propanol and Poly(vinyl imidazole-co-vinyltrimethoxysilane)	Room temperature under ambient conditions for 3–4 h (No thermal annealing)	12,000 on glass	[119]
Cu@Ag NPs/11.7 nm	1-amino-2-propanol (AmIP)	Room temperature (No thermal annealing)/ligand exchange from OAM to AmIP and reduction-assisted sintering with immersionin a NaBH_4_ solution	36.3 on PET	[120]
Cu NPs/11.9 nm	Oleylamine (OAM)	Room temperature (No thermal annealing)/reduction-assisted sintering with OAM removing by HCOOH and reduction by NaBH_4_	16.93 on PET	[121]

**Table 6 nanomaterials-10-01689-t006:** Summary of the formulation designs in Cu-based mixed inks/pastes.

Conductive Sources	Protective Agents	Sintering Process	Resistivity (μΩ·cm)	Ref.
1. Cu NPs (60.8 nm) 2. Cu flakes (9.3 μm)	Decanoic acid	120 °C/N_2_ (95%)–H_2_ (5%)/180 min	29 on PET (NP:Flake =80 wt%:20 wt%)	[124]
1. Cu NPs (3–5 nm) 2. Cu particles (Ps) (300 nm)	1-amino-2-propanol (AmIP)	150 °C/in air/10 s 120 °C/N_2_/60 min	55 on PI (NPs:Ps = 1:3) (mass ratio) 8.4 on PI (NPs:Ps = 1:3) (mass ratio)	[125]
1. Cu NPs (93 nm) 2. Cu flakes (3 μm)	Oleic acid	IPL	11.4 on PI (NP:Flake = 4:6) (mass ratio)	[126]
1. Cu NPs (100 nm) 2. Cu particles (Ps) (1.8 μm)	N.A.	IPL	6.94 on PI (NPs:Ps = 1:1) (mass ratio)	[127]
1. Cu NPs (100 nm) 2. Cu particles (Ps) (6–8 μm)	N.A.	IPL	6.16 on PI (NPs:Ps = 1:1) (mass ratio)	[128]
1. Cu NPs (180 nm) 2. Cu particles (Ps) (2 μm)	N.A.	IPL	5.94 on PI (NPs:Ps = 1:1) (mass ratio)	[129]
1.Cu NPs (150 nm) 2.SnNPs (100–300 nm)	PVP	IPL	14 Ω/sq on metal-mesh (CuNPs:SnNPs = 2:1) (mass ratio)	[130]
1.Cu particles (Ps) (800 nm) 2. CuF–AmIP complex	N.A.	100 °C/N_2_/60 min	900 on alumina (complex: Ps = 1:6) (molar ratio)	[133]
1.Cu particles (Ps) (400 nm) 2. CuF–AmIP complex	N.A.	120 °C/N_2_/30 min	32 on alumina (70 wt% Cu particle)	[134]
1.Cu particles (Ps) (154 nm) 2. CuF-AmIP complex	poly(propylenecarbonate)	100 °C/N_2_/60 min	88 on alumina (complex: Ps = 1:2) (molar ratio)	[135]
1.Cu particles (Ps) (700 nm) 2.CuF–2-amino-2-methyl-1-propanol (AMP) complex	N.A.	140 °C/N_2_/15 min	11.3 on PET (complex: Ps = 3: 1) (mass ratio)	[136]
1.Cu NPs (60–100 nm) 2.CuF–3-dimethylamino-2-propanediol (DEAPD) complex	N.A.	200 °C/N_2_/60 min	18 on PI (NPs: 0.15 g, complex:0.092 g)	[137]
1. AMP + Cu(OH)_2_ + formic acid 2. Cu flake (9.5 μm)	Decanoic acid	200 °C/N_2_/3 min	21 on PI [Cu(OH)_2_/Formic acid ratio of 0.875 (22.7 wt% copper) and Cu flake 7.3 wt%]	[138]
1.Cu flake (1–2 mm) 2. CuF–AmIP complex	N.A.	150 °C/N_2_/60 min	62 on cellulose paper (Flake: complex = 3:1) (mass ratio)	[139]
1. Formic acid-treated Cu flake (1–2 μm) 2. CuF–AmIP complex	N.A.	100 °C/in air/15 s	70 on cellulose paper (Flake: complex = 3:1) (mass ratio)	[140]
1.Cu NPs (10–70 nm) 2.Cu nitrate trihydrate	PVP	IPL	27.3 on PI (30 wt% Cu nitrate)	[141]
1.Cu particles (Ps) (250 nm) 2.CuF–AMP complex	N.A.	140 °C in N_2_ for 10 min, and IPL	7.2, 26.5 and 15.9 on PI, PET, and PEN (5 g of Cu formate tetrahydrate,3.95 g of AMP, and 2.98 g of Cu particles)	[142]
1.Cu particles (Ps) (350 nm) 2.CuF–AMP complex	N.A.	140 °C in N_2_ for 10 min, and IPL	5.8 on PI (10 g of Cu formate tetrahydrate, 7.9 g of AMP, and 5.96 g of Cu particles)	[143]
1.Cu NPs 2.CuF–DEAPD complex	N.A.	100–135 °C for 20 min, and IPL	20−38 on 70−550 μm trace widths on PI [Cu(II) formate (40.20%), DEAPD (43.70%), CuNP (0.40%)]	[144]
1.Cu particles (Ps) (350 nm) 2. Ag-amino complex	N.A.	140 °C in N_2_ for 10 min, and IPL	3.4 on PI [Ag(I) β-ketocarboxylate:2-ethylhexylamine = 1:2 (molar ratio), Cu:Ag =80 wt%: 20 wt%]	[146]
1.Cu NWs (150 nm in diameter, 1−2 μm in length) 2. Cu NPs (20–50 nm)	PVP	IPL	22.77 on PI (CnNP:Cu NW = 95 wt%:5 wt%)	[150]
1.Cu NWs (150 nm in diameter, 1−2 μm in length) 2. CuF–2-ethylhexylamine complex	Octadecyl amine	140 °C/N_2_/30 min	14.9 on PET (10 wt% Cu NW)	[151]
1.MWCNT (length = 200 μm) 2. Cu NPs (30–50 nm)	PVP	IPL	7.86 on PI (0.5 wt% CNT)	[152]
1 SWCNT (length = 200 μm) 2. Cu formate particles (50–500 nm)	PVP	IPL	0.104 S om PI	[153]
1.MWCNT 2. CuF–AMP complex	Octyl amine	140 °C/N_2_/30 min	25.31 on PET (1 wt% MWCNT)	[154]
1.Cu NPs (10–70 nm) 2.Cu nitrate trihydrate	PVP 3 wt% APS	IPL	7.6 on PI (30 wt% Cu NPs)	[155]
Cu ion/Formic acid/ammonia/citric acid	N.A.	IPL	2.3 on SAM surface treatment PI	[156]

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
