# Peer review of "Surface and Interface Designs in Copper-Based Conductive Inks for Printed/Flexible Electronics"

_nanomaterials, 2020, doi:10.3390/nano10091689_

Round 1

Reviewer 1 Report

In this work, "Surface and interface designs in copper-based conductive inks for printed/flexible electronics", the authors summarized the surface and interface designs for Cu-based conductive inks/pastes, where the strategies for the oxidation resistance of Cu and the low-temperature sintering are applied to generate highly conductive Cu patterns/electrodes on flexible substrates. Overall, this Review manuscript has a strong potential for a second review after applying the issues and addressing the shortcomings listed below:

1-The authors should polish/revise some grammatical mistakes and typos along the manuscript. For instance, ‘Surface Desigins…’. From now on, I am not mentioning the remaining grammatical mistakes/typos along the manuscript. I invite the authors to read their manuscript carefully and make the required changes where necessary.

2-Revise the following statement: ‘Cu has high intrinsic…’. From now on, I am not mentioning the remaining grammatical mistakes/typos along the manuscript. I invite the authors to read their manuscript carefully and make the required changes where necessary.

3-Try to enhance the resolution of the Figures, especially Figures 4 and 11.

4-In the Introduction section, while discussing some of the features of Au and application areas, the following works should also be considered and cited, to give a more general view to the possible readers of the Review: [(i) Monolithic metal dimer-on-film structure: new plasmonic properties introduced by the underlying metal, Nano Letters 20, 2087-2093 (2020); (ii) The role of Ge2Sb2Te5 in enhancing the performance of functional plasmonic devices, Materials Today Physics 12, 100178 (2020)].

Author Response

Dear Reviewer,
First of all, we would like to thank you for your efforts in referring our manuscript. We have taken into consideration all of your comments and made the changes. These are listed hereafter. The original reviewers’ comments are copied below in italics, our responses are provided below those of the reviewers, and changes to the manuscript are highlighted in their respective colors.

Prof. Hideya Kawasaki

Reviewer: 1
In this work, "Surface and interface designs in copper-based conductive inks for printed/flexible electronics", the authors summarized the surface and interface designs for Cu-based conductive inks/pastes, where the strategies for the oxidation resistance of Cu and the low-temperature sintering are applied to generate highly conductive Cu patterns/electrodes on flexible substrates. Overall, this Review manuscript has a strong potential for a second review after applying the issues.
Reply: We thank the reviewer for giving useful comments.

1) The authors should polish/revise some grammatical mistakes and typos along the manuscript. For instance, ‘Surface Desigins…’. From now on, I am not mentioning the remaining grammatical mistakes/typos along the manuscript. I invite the authors to read their manuscript carefully and make the required changes where necessary. 2) Revise the following statement: ‘Cu has high intrinsic…’. From now on, I am not mentioning the remaining grammatical mistakes/typos along the manuscript. I invite the authors to read their manuscript carefully and make the required changes where necessary.
Reply: Thank you so much careful reading our manuscript. We carefully revised grammatical mistakes and typos. 3) Try to enhance the resolution of the Figures, especially Figures 4 and 11.
Reply: Thank you for your suggestion. We enhanced the resolution of the Figures.

4) In the Introduction section, while discussing some of the features of Au and application areas, the following works should also be considered and cited, to give a more general view to the possible readers of the Review: [(i) Monolithic metal dimer-on-film structure: new plasmonic properties introduced by the underlying metal, Nano Letters 20, 2087-2093 (2020); (ii) The role of Ge2Sb2Te5 in enhancing the performance of functional plasmonic devices, Materials Today Physics 12, 100178 (2020)]s Today Physics 12, 100178 (2020)].
Reply: Thank you for your information on the papers to give a more general view. We cited the papers the reviewer suggested as ref. 11 and 12.

Reviewer 2 Report

This work is well presented, has very valuable data, and covers the literature very well.

Here are my main observations that the authors should take into account:

1. The paragraph 4.3.2 entitled "Carbon shell", could be more extended.

2. If possible, the figure quality should be improved (e.g. Figures 6b or 13a, c or 19e).

3. Please, correct the following typos errors

                -p.26, line 973: the large volume sheinkage

                -p. 6, line 217 : Surface Desigins by Surface Protective Layers

4. Please, check the meaning of lines 123-124 in p. 3 (In line 123 replace micro with nano).

5. Where reference is made to text in printing inks then the majority of the selected citations are based on inks applied to the screen printing method and not for example to gravure or the flexography. I wonder if there is a specific reason. In this case some justification could be added.

Author Response

Dear Reviewer,
First of all, we would like to thank you for your efforts in referring our manuscript. We have taken into consideration all of your comments and made the changes. These are listed hereafter. The original reviewers’ comments are copied below in italics, our responses are provided below those of the reviewers, and changes to the manuscript are highlighted in their respective colors.

Prof. Hideya Kawasaki

Reviewer: 2

In the present work the authors give an exhaustive review on Cu-based inks for printed/flexible electronics application. In particular, they focus on disadvantages of Cu-based inks/pastes, mainly related to Cu instability against oxidation under ambient conditions and its tendency to form insulating layers of Cu oxide. The review is structured in seven sections where the authors summarize the surface and interface designs for Cu-based conductive inks/pastes, in which the strategies for the oxidation resistance of Cu and the low-temperature sintering are applied to produce highly conductive Cu patterns/electrodes on flexible substrates. Recently developed Cu-based mixed inks/pastes are also described together with a brief discussion on the adhesion enhancement of sintered Cu films on flexible subtrates. Finally the authors offer their perspectives of Cu-based inks/pastes for future efforts. The present review is well organized and all the sections are well balanced. The treated arguments are well described and supported by adequate references. I suggest it for publication on Nanomaterials only after some minor revisions.

Reply: We thank the reviewer for their positive comments, which are insightful for us to improve our manuscript.

1. All tables presented in the whole review should be described in the main text.
Reply: Thank you for your suggestion. We described all tables presented in the whole review in the main text.

2. The acronyms present in the text should be firstly explained to be understandable to Non-specialists, i.e. lines 282 (HTHH), 351 (FOM), 439 (OAM), 494 (AA), 621 (PAA), 645 (CTAB), 724 (OAM), 951 (folic acid (Met(OH)), 1078 (PCC), 1238 (CNTs).
Reply: I am sorry that we did not explain these acronyms. We added the explanation of these acronyms in the revised manuscript.

3. Line 217 Design, - Line 268 17 and 5.7 μΩ cm - Line 1097 were.
Reply: Thank you so much careful reading our manuscript. We carefully revised grammatical mistakes and typos.

Author Response

Dear Reviewer,
First of all, we would like to thank you for your efforts in referring our manuscript. We have taken into consideration all of your comments and made the changes. These are listed hereafter. The original reviewers’ comments are copied below in italics, our responses are provided below those of the reviewers, and changes to the manuscript are highlighted in their respective colors.

Prof. Hideya Kawasaki

Reviewer: 3

In the present work the authors give an exhaustive review on Cu-based inks for printed/flexible electronics application. In particular, they focus on disadvantages of Cu-based inks/pastes, mainly related to Cu instability against oxidation under ambient conditions and its tendency to form insulating layers of Cu oxide. The review is structured in seven sections where the authors summarize the surface and interface designs for Cu-based conductive inks/pastes, in which the strategies for the oxidation resistance of Cu and the low-temperature sintering are applied to produce highly conductive Cu patterns/electrodes on flexible substrates. Recently developed Cu-based mixed inks/pastes are also described together with a brief discussion on the adhesion enhancement of sintered Cu films on flexible subtrates. Finally the authors offer their perspectives of Cu-based inks/pastes for future efforts. The present review is well organized and all the sections are well balanced. The treated arguments are well described and supported by adequate references. I suggest it for publication on Nanomaterials only after some minor revisions.

Reply: We thank the reviewer for their positive comments, which are insightful for us to improve our manuscript.

1. All tables presented in the whole review should be described in the main text.
Reply: Thank you for your suggestion. We described all tables presented in the whole review in the main text.

2. The acronyms present in the text should be firstly explained to be understandable to Non-specialists, i.e. lines 282 (HTHH), 351 (FOM), 439 (OAM), 494 (AA), 621 (PAA), 645 (CTAB), 724 (OAM), 951 (folic acid (Met(OH)), 1078 (PCC), 1238 (CNTs).
Reply: I am sorry that we did not explain these acronyms. We added the explanation of these acronyms in the revised manuscript.

3. Line 217 Design, - Line 268 17 and 5.7 μΩ cm - Line 1097 were.
Reply: Thank you so much careful reading our manuscript. We carefully revised grammatical mistakes and typos.

Author Response

Dear Reviewer,
First of all, we would like to thank you for your efforts in referring our manuscript. We have taken into consideration all of your comments and made the changes. These are listed hereafter. The original reviewers’ comments are copied below in italics, our responses are provided below those of the reviewers, and changes to the manuscript are highlighted in their respective colors.

Prof. Hideya Kawasaki

Reviewer: 4
The manuscript nanomaterials-915968 provides a comprehensive overview of recent progress on the application of copper-based conductive inks for printed/flexible electronics. The review is well written and very well organized. It covers the different types of Cu-based inks including all aspects of surface designs, from small organic ligands to core shell structures. The work also considers surface activation as well as adhesion enhancement strategies. The contents are interesting and can be useful for the audience of the journal in order to inspire the design of novel copper-based ink/pastes for printed electronics. For these reasons, I recommend the publication of the manuscript as a review in the Nanomaterials journal. I would only like to comment a couple of issues that the authors might consider to revise:

Reply: We thank the reviewer for their positive comments

1.Although the review is focused on the discussion of Cu-based inks/pastes for flexible electronics applications, a brief outline of potential developments in the analytical biosensing field could be included in the conclusions and outlook section.

Reply: Thank you for your suggestion. We added the outlooks for a brief outline of potential developments in the analytical biosensing field(page 42, line 1499-1526). The related references (ref. 160-167) are newly added in the revised manuscript.

2. Typo errors: line 217, 4. Surface Desigins.

Reply: Thank you so much careful reading our manuscript. We carefully revised grammatical mistakes and typos in the revised manuscript.

Reviewer 5 Report

Report on manuscript nanomaterials-915968 Tomotoshi:

In the manuscript entitled "Surface and interface designs in copper-based conductive inks for printed/flexible electronics," Tomotoshi and Kawasaki provided a comprehensive review of recent progress on the surface and interface designs for Cu-based conductive inks and pastes. This review is written well and the content is organized in an easy accessible way. By overviewing recent major achievements from different aspects, the authors also made necessary comparisons between different approaches. In the end, the authors further offered their perspectives on the topic.

After carefully reading the manuscript, I found this review is very interesting and the authors did an excellent job on reviewing the critical developments in the field. I expect this review would impact on the field.

I do not find serious issues in the content. The main issue I noticed is the quality of some figures. In particular, some subfigures in Figures 3-13 and 15-21 are hard to read due to the poor quality of the images. For example, in Figs. 3(a) and (b) the displayed digits are hard to recognize. Inside Figs. 4 (a)-(c), the English is very difficult to read. There are also some typos in the content. Please consider to correct them.

In short, I would like to recommend it for publication in Nanomaterials. As the identified issues are nontechnical, no further review is necessary.

Author Response

Dear Reviewer,
First of all, we would like to thank you for your efforts in referring our manuscript. We have taken into consideration all of your comments and made the changes. These are listed hereafter. The original reviewers’ comments are copied below in italics, our responses are provided below those of the reviewers, and changes to the manuscript are highlighted in their respective colors.

Prof. Hideya Kawasaki

Reviewer 5
In the manuscript entitled "Surface and interface designs in copper-based conductive inks for printed/flexible electronics," Tomotoshi and Kawasaki provided a comprehensive review of recent progress on the surface and interface designs for Cu-based conductive inks and pastes. This review is written well and the content is organized in an easy accessible way. By overviewing recent major achievements from different aspects, the authors also made necessary comparisons between different approaches. In the end, the authors further offered their perspectives on the topic. After carefully reading the manuscript, I found this review is very interesting and the authors did an excellent job on reviewing the critical developments in the field. I expect this review would impact on the field.
I do not find serious issues in the content. The main issue I noticed is the quality of some figures. In particular, some subfigures in Figures 3-13 and 15-21 are hard to read due to the poor quality of the images. For example, in Figs. 3(a) and (b) the displayed digits are hard to recognize. Inside Figs. 4 (a)-(c), the English is very difficult to read. There are also some typos in the content. Please consider to correct them. In short, I would like to recommend it for publication in Nanomaterials. As the identified issues are nontechnical, no further review is necessary.

Reply: We thank the reviewer for their positive comments. We enhanced the resolution of the Figures.

Reviewer 6 Report

This review comprehensively described the copper-based electrodes for flexible electronic devices. The authors provided the fabrication process, properties, and applications of copper-based flexible electrodes. I believe this review is very well organized and has no critical defects. However, some contents should be included in revision process. The authors should add the outlooks for the future such as development tendency and potential applications of flexible metal-based electrodes. After revision process with above-mentioned comments, I think that this review do satisfy the scientific/technical rigor needed for this journal and is suitable for publication in this journal.

Author Response

Dear Reviewer,
First of all, we would like to thank you for your efforts in referring our manuscript. We have taken into consideration all of your comments and made the changes. These are listed hereafter. The original reviewers’ comments are copied below in italics, our responses are provided below those of the reviewers, and changes to the manuscript are highlighted in their respective colors.

Prof. Hideya Kawasaki

Reviewer 6
This review comprehensively described the copper-based electrodes for flexible electronic devices. The authors provided the fabrication process, properties, and applications of copper-based flexible electrodes. I believe this review is very well organized and has no critical defects. However, some contents should be included in revision process. The authors should add the outlooks for the future such as development tendency and potential applications of flexible metal-based electrodes. After revision process with above-mentioned comments, I think that this review do satisfy the scientific/technical rigor needed for this journal and is suitable for publication in this journal.

Reply: Thank you for your suggestion. We added the outlooks for a brief outline of potential developments in the analytical biosensing field(page 42, line 1499-1526). The related references (ref. 160-167) are newly added in the revised manuscript.

Round 2

Reviewer 1 Report

In its current form, the revised manuscript is suitable for publication.